# FLOW DIVERSE AND EFFICIENT: LEARNING MOMENTUM FLOW MATCHING VIA STOCHASTIC VELOCITY FIELD SAMPLING

## ABSTRACT

Recently, the rectified flow (RF) has emerged as the new state-of-the-art among flow-based diffusion models due to its high efficiency advantage in straight path sampling, especially with the amazing images generated by a series of RF models such as *Flux 1.0* and *SD 3.0*. Although a straight-line connection between the noisy and natural data distributions is intuitive, fast, and easy to optimize, it still inevitably leads to: *1) Diversity concerns*, which arise since straight-line paths only cover a fairly restricted sampling space. *2) Multi-scale noise modeling concerns*, since the straight line flow only needs to optimize the constant velocity field $v$ between the two distributions $\pi_0$ and $\pi_1$. In this work, we present Discretized-RF, a new family of rectified flow (also called momentum flow matching models since they refer to the previous velocity component and the random velocity component in each diffusion step), which discretizes the straight path into a series of variable velocity field sub-paths (namely *"momentum fields"*) to expand the search space, especially when close to the distribution $p_{\text{noise}}$. Different from the previous case where noise is directly superimposed on $x$, we introduce noise on the velocity $v$ of the sub-path to change its direction in order to improve the diversity and multi-scale noise modeling abilities. Experimental results on several representative datasets demonstrate that learning momentum flow matching by sampling random velocity fields will produce trajectories that are both diverse and efficient, and can consistently generate high-quality and diverse results.

## 1 INTRODUCTION

Flow-based diffusion models (Lipman et al.; Bartosh et al., 2024; Luo et al., 2024; Liu et al., 2023b) have recently attracted widespread attention, which generate a wide variety of realistic natural images from pure noise by modeling trajectories from noise distributions to data distributions. As a milestone work, the most popular flow models currently are the rectified flow (RF) models (Liu et al., 2023a) built upon straight-line trajectories, which significantly improves their sampling efficiency by establishing the shortest straight-line connection between the noise distribution $\pi_0$ and the data distribution $\pi_1$. Furthermore, the models can be easily optimized by directly calibrating this straight-line trajectory $dx_t/dt = v_\theta$ at a constant rate $x_1 - x_0$. Due to its high sampling efficiency (even enabling one-step diffusion generation), RF is also considered to be one of the fastest flow-based optimal transport models.

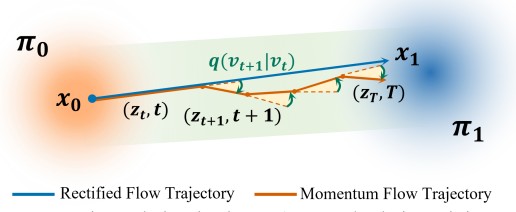

Figure 1: Graphical momentum flow trajectories. **Momentum Flow (orange)** *vs.* **Rectified Flow (blue)**.

Though remarkable success has been witnessed, RF still suffers from limitations in *diversity* and *multi-scale noise modeling*. Specifically, *1) Diversity concerns*, which arise since the straight-line path can only cover a fairly restricted sampling space. *2) Multi-scale noise modeling concerns*, since the straight-line flow only needs to directly optimize the constant velocity field $v_\theta \to (x_1 - x_0)$ between the two distributions $\pi_0$ and $\pi_1$, instead of considering multi-scale progressive denoising. At the other extreme, the diffusion probability models (*e.g.,* DDPM) based on fluctuation trajectories

have extremely strong diversity and multi-scale noise modeling capabilities but face the challenge of training- and sampling-efficiency because they require a large number of time steps to sample, and each step should be optimized iteratively to achieve high-fidelity modeling of the reverse trajectory. In this work, to strike a balance and take into account both efficiency and diversity (especially the potential diversity when close to noise distribution $\boldsymbol{\pi}_1$), we propose a Discretized-RF model, also known as the momentum flow matching (**MFM**) model. For clarity, we first give a unified definition of the flow transport problem and then introduce our momentum flow transport.

**Flow Transport Problem Definition.** *Given empirical observations of two distributions $\boldsymbol{x}_0 \sim \boldsymbol{\pi}_0$ (real data distribution) and $\boldsymbol{x}_1 \sim \boldsymbol{\pi}_1$ (noise distribution) on $\mathbb{R}^d$, define a forward transport trajectory $T_o : \mathbb{R}^d \to \mathbb{R}^d$ that satisfies $\boldsymbol{x}_1 := T_o(\boldsymbol{x}_0) \sim \boldsymbol{\pi}_1$ when $\boldsymbol{x}_0 \sim \boldsymbol{\pi}_0$. At the same time, the forward flow transport should own the estimable property of the reverse solution trajectory, that is, $\boldsymbol{x}_0 := \tilde{T}_\theta(\boldsymbol{x}_1) \sim \boldsymbol{\pi}_0$ when $\boldsymbol{x}_1 \sim \boldsymbol{\pi}_1$, which requires the trajectory to be continuous and tractable.*

**Momentum Flow Transport (Discretized-RF).** *Given the shortest optimal transport $d\boldsymbol{x}_t/dt = \boldsymbol{v}$ (straight-line trajectory) at a constant rate $\boldsymbol{v} = \boldsymbol{x}_1 - \boldsymbol{x}_0$ and a series of discretized anchor points $\{\boldsymbol{z}_1, \cdots, \boldsymbol{z}_{T-1}\}$, define a segmented straight-line trajectory $T_{\boldsymbol{x}_0 \mapsto \boldsymbol{x}_1} = \{\boldsymbol{x}_0, \boldsymbol{z}_1, \cdots, \boldsymbol{z}_{T-1}, \boldsymbol{x}_1\}$ that satisfies $d\boldsymbol{z}_t/dt = \boldsymbol{v}_t$ and $\boldsymbol{v}_t = \sqrt{\gamma}\boldsymbol{v}_{t-1} + \sqrt{(1-\gamma)}\boldsymbol{\epsilon}_t, \boldsymbol{\epsilon}_t \sim \mathcal{N}(0, \boldsymbol{I})$. Meanwhile, the endpoint transport of this momentum flow trajectory are respectively defined as: $T_{\boldsymbol{x}_0 \mapsto \boldsymbol{z}_1} : d\boldsymbol{z}_t/dt = \boldsymbol{v}_0$ (initialized by $\boldsymbol{x}_1 - \boldsymbol{x}_0$) and $T_{\boldsymbol{z}_{T-1} \mapsto \boldsymbol{x}_1} : d\boldsymbol{z}_t/dt = \boldsymbol{\epsilon}, \boldsymbol{\epsilon} \sim \mathcal{N}(0, \boldsymbol{I})$. The momentum flow ensures that the velocity is Gaussian divergent when approaching $\boldsymbol{\pi}_1$, while the velocity is more deterministic and faster when approaching $\boldsymbol{\pi}_0$. Note that the acceleration $\boldsymbol{\epsilon}$ follows the same Gaussian distribution $\mathcal{N}(0, \boldsymbol{I})$ and can therefore be easily estimated by the neural model $\boldsymbol{\epsilon}_\theta$ to obtain a learnable and tractable inverse transport trajectory $\tilde{T}_{\boldsymbol{x}_1 \mapsto \boldsymbol{x}_0; \theta} = \{\boldsymbol{x}_1, \boldsymbol{z}_{T-1; \theta}, \cdots, \boldsymbol{z}_{1; \theta}, \boldsymbol{x}_{0; \theta}\}$.*

Beyond image generation, the challenge of balancing efficiency and diversity is even more pronounced when generating 3D geometric structures, where the data lies on inherently non-Euclidean manifolds instead of the common Euclidean manifold $\mathbb{R}^d$. Consequently, to further demonstrate the scope and applicability of our momentum flow, we extend it to the **Special Euclidean group** SE(3) for protein backbone generation. In this context, each amino acid residue is represented by a *frame* (i.e., 3D rigid body) in SE(3), parameterizing its spatial orientation and position. This extension is profoundly advantageous: by leveraging the Lie algebra $\mathfrak{se}(3)$, which is the tangent space of SE(3) and linearly isomorphic to $\mathbb{R}^6$, we transform the complex nonlinear manifold of protein structures into a vector space where rotations and translations are seamlessly unified within a single stochastic momentum field, achieving diverse and efficient frame sampling without expensive SE(3) geodesic calculations.

The goal of this work is to extend the constant velocity field model to the acceleration field model by learning the momentum flow matching via stochastic velocity field sampling, so as to finally derive a compromise transport path with both speed and diversity. Main contributions are summarized below:

- **A momentum-driven flow model for reasonable diversity-efficiency trade-off:** Is the straighter the flow, the better? Unlike rectified flows that are modeled on a straight-line trajectory or diffusion models that adopt completely stochastic paths, our momentum flow discretizes the straight path into a series of variable velocity field sub-paths. This makes the trajectory more deterministic (*efficient*) near the data distribution and more random (*diverse*) near the noise distribution, thus achieving a proper balance between diversity and efficiency without sacrificing straight-path advantages.

- **Segmented straight-line sampling for multi-scale noise modeling:** Our proposed Discretized-RF solution trajectory (i.e., *segmented flow trajectory* where each small segment is a straight line) is a better approximation of multi-scale noise-adding and denoising. It is easier to optimize than the stochastic differential equation (i.e., *fluctuation flow trajectory*) and can better model multi-scale noise than the constant velocity field differential equation (i.e., *rectified flow trajectory*).

- **Superior performance on multiple image datasets**: Extensive experiments show that our Momentum Flow achieves competitive FID and recall scores with substantially fewer denoising steps. Specifically, on multiple image datasets, including CIFAR-10 (Krizhevsky, 2009), CelebA-HQ (Karras et al., 2018) and ImageNet (Deng et al., 2009), it consistently matches or even outperforms the performance of Rectified Flow while requiring only half the number of sampling steps.

- **High adaptability to SE(3) for protein generation:** By unifying rotations and translations in a single stochastic momentum field via $\mathfrak{se}(3)$-$\mathbb{R}^6$ isomorphism, our MFM enables efficient and diverse *frame* sampling, demonstrating its better adaptability to non-Euclidean manifolds than RF.

## 2 RELATED WORK

**Diffusion Models: High Diversity at the Cost of Efficiency.** Diffusion models (Song & Ermon, 2019; Ho et al., 2020; Song et al., 2020b; Nichol & Dhariwal, 2021; Kawar et al., 2022; Ma et al., 2024d; Ma et al.; 2025; 2024b) have emerged as a powerful class of generative models, known for their impressive sample diversity. However, their stochastic diffusion trajectories typically require hundreds or thousands of sampling steps, leading to significant computational costs. To overcome this inefficiency, researchers have proposed some optimization methods along two primary directions: sampling acceleration strategies (Liu et al., 2022a; Salimans & Ho, 2022; Gonzalez et al., 2023; Meng et al., 2023; Song et al., 2023; Sauer et al., 2024; Xu et al., 2024) and model architecture improvements (Li et al., 2023; Zhao et al., 2024; Xu et al., 2024; Li et al., 2024a; Ma et al., 2024c). For instance, DDIM (Song et al., 2020a) introduces a non-Markovian reverse process that decouples temporal dependencies, substantially reducing the number of sampling steps. DeepCache (Ma et al., 2024a) accelerates inference by caching and retrieving features across adjacent denoising stages to avoid redundant computations. On the architectural side, some works enhance model efficiency by employing custom multi-decoder U-Net designs that combine time-specific decoders with a shared encoder (Zhang et al., 2024), or by enabling parallel decoder execution to speed up the denoising process (Li et al., 2024b). Moreover, in the field of protein backbone generation, FrameDiff (Yim et al., 2023b) develops a SE(3)-invariant diffusion model on $SE(3)^N$ for protein modelling, thereby generating designable, novel and diverse monomers beyond the Protein Data Bank (PDB) (Berman et al., 2000) without relying on a pretrained protein structure prediction network. Despite these advances, diffusion-based models still depend on curved stochastic paths, which remain inherently more expensive to compute than deterministic or straight-path methods. As a result, the fundamental trade-off remains: high sample diversity comes at the expense of computational efficiency.

**Rectified Flows: Faster Sampling Meets Less Diversity.** Rectified flow models (Liu, 2022; Liu et al., 2023a;b; Wang et al., 2024a; Zhu et al., 2024b; Gat et al., 2024) significantly improve sampling efficiency over diffusion models by optimizing straight-line trajectories in probability space. However, their deterministic and straight sampling paths fundamentally limit their diversity. To solve this issue, various techniques have been proposed to enhance sample diversity while maintaining efficiency. Some methods focus on optimizing noise sampling techniques (Yan et al., 2024; Wang et al., 2024c; Liu et al., 2024), such as training on perceptually relevant noise scales (Esser et al., 2024) or sampling from multi-modal flow directions (Guo & Schwing, 2025). Other efforts aim to improve generation quality (Lee et al., 2024; Li et al., 2024c; Dalva et al., 2024) include applying flow matching in the latent space of pretrained autoencoders (Dao et al., 2023), mitigating numerical errors in the ODE-solving process (Wang et al., 2024b) or introducing posterior-mean-based optimal estimators (Ohayon et al., 2025). Moreover, some protein-related methods (Yim et al., 2023a; Campbell et al., 2024; Lin et al., 2024) utilize RF for protein modelling, achieving speedup during frame sampling. However, the trade-off between sampling speed and diversity persists, motivating the development of adaptive flow-based methods that preserve computational efficiency while enhancing sampling diversity.

Unlike previous methods, our momentum flow matching model introduces a momentum field into the forward process, where multi-scale noise dynamically adjusts the trajectory directions to promote sampling diversity. To improve computational efficiency, the reverse trajectory is discretized into multiple sub-paths, each optimized via rectified flow. As a result, our model retains the fast sampling speed of rectified flow while recovering much of the sample diversity achieved by diffusion models.

## 3 METHOD

In this section, we propose Momentum Flow Transport, a novel flow-based diffusion model family that aims to achieve an effective balance between diversity and efficiency via a brand-new momentum flow matching technique in Sec. 3.1. Momentum Flow is a dynamically compromise approximation of multi-scale noise-adding (or de-noising) between a straight line and a fluctuating line by combinating: 1) *fluctuation flow trajectory (**close to** $x_1$)* for diversity and 2) *rectified flow trajectory (**close to** $x_0$)* for efficiency. We then further introduce the momentum-guided forward process in Sec. 3.2, the acceleration fields-driven reverse process in Sec. 3.3 and its extension to SE(3) in Sec. 3.4.

### 3.1 MOMENTUM FLOW MATCHING

**Optimal Transport (OT).** The optimization problem from noise distribution $\pi_1$ to data distribution $\pi_0$ can be regarded as an optimal transport (OT) problem. Since it is extremely difficult to directly

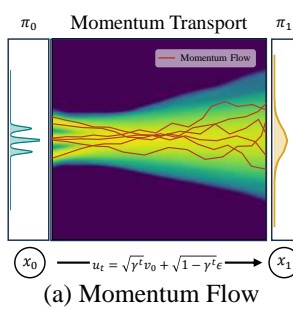 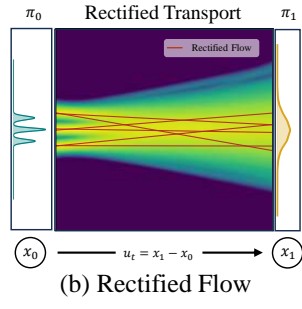 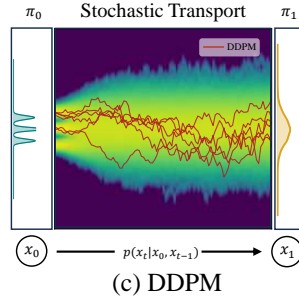

(a) Momentum Flow      (b) Rectified Flow      (c) DDPM

Figure 2: Overview of Momentum Flow. Compared with Rectified Flow (Liu et al., 2023a) (*Efficiency*-OT) and DDPM (Ho et al., 2020) (*Diversity*-OT), the momentum flow tends to explore diversity when close to noise distribution $\boldsymbol{\pi}_1$, and tends to focus on efficiency when close to data distribution $\boldsymbol{\pi}_0$.

solve the trajectory from $\boldsymbol{\pi}_1$ to $\boldsymbol{\pi}_0$, recent flow-based methods (Lipman et al.; Liu et al., 2023a) usually first give a tractable forward trajectory $T_o$ to transport any $\boldsymbol{x}_0 \sim \boldsymbol{\pi}_0$ to $\boldsymbol{x}_1 \sim \mathcal{N}(0, \boldsymbol{I})$ (approximation of $\boldsymbol{\pi}_1$), and then solve the posterior $p(\boldsymbol{\pi}_0|\boldsymbol{\pi}_1) = \tilde{T}_\theta(\boldsymbol{\pi}_1)$ via a flow-matching trajectory estimator $\tilde{T}_\theta$,

$$\boldsymbol{\pi}_1 = T_o(\boldsymbol{\pi}_0) = \int d\boldsymbol{z}_t T_o\left(\boldsymbol{\pi}_1 \mid \boldsymbol{z}_t\right) \boldsymbol{\pi}\left(\boldsymbol{z}_t\right), \ \ \boldsymbol{\pi}_0 = \tilde{T}_\theta(\boldsymbol{\pi}_1) = \int d\boldsymbol{z}_{(0:T)} \boldsymbol{\pi}\left(\boldsymbol{z}_T\right) \prod_{t=1}^{T} p\left(\boldsymbol{z}_{t-1} \mid \boldsymbol{z}_t\right).$$

(1)

**Stochastic Transport (*Diversity*-OT) and Rectified Transport (*Efficiency*-OT).** Stochastic Transport (Ho et al., 2020; Song et al., 2020a) and Rectified Transport (Liu et al., 2023a;b) are two common optimal transport methods, which are respectively known for their high sampling quality (*diversity*) and fast sampling speed (*efficiency*). However, they both struggle with the balance between efficiency and diversity, either relying on overly divergent sampling steps (trajectory variance $\boldsymbol{\beta}_T \to \infty$) or predefined straight trajectories ($\boldsymbol{\beta}_T = 0$). Our work aims to find a balanced trajectory $T_o$ in terms of optimal efficiency and optimal diversity so that the trajectory variance tends to $0$ when close to data distribution $\boldsymbol{\pi}_0$ (for *efficiency*) and tends to $\infty$ when close to noise distribution $\boldsymbol{\pi}_1$ (for *diversity*).

**Momentum Field (Acceleration Field).** In order to find a balanced trajectory, we introduce the momentum field. That is a variable velocity field referring to the previous velocity component and the random velocity component in each diffusion step. Let $\boldsymbol{\nu} = \{\boldsymbol{v}_t\}_0^{T-1}$ represent the momentum field (for guiding $\boldsymbol{x}_0$ to $\boldsymbol{x}_1$), $\boldsymbol{v}_t$ denote the velocity vector from time $t$ to time $t+1$, we have:

$$\frac{d\boldsymbol{z}_t}{dt} = \boldsymbol{v}_t, \ \ \boldsymbol{v}_t = \begin{cases} \beta(\boldsymbol{\epsilon}_0 - \boldsymbol{x}_0) & \text{if } t = 0 \\ \sqrt{\gamma_t}\boldsymbol{v}_{t-1} + \sqrt{1 - \gamma_t}\beta\boldsymbol{\epsilon}_t & \text{if } 0 < t < T \\ \beta\boldsymbol{\epsilon}_T & \text{if } t = T. \end{cases}$$

(2)

Here $\boldsymbol{z}_t \sim \boldsymbol{\pi}(\boldsymbol{z}_t)$ is the middle noise-perturbed distribution during the forward diffusion process and $\{\gamma_t\}_1^{T-1}$ is the momentum decay coefficient, which can be chosen as a constant $\gamma$ ($\gamma < 1$) or a positive decreasing series. We choose the former in our work. For convenience, $\beta$ denotes the normalization coefficient $\beta := (\sqrt{\gamma} - 1)/(\sqrt{\gamma^T} - 1)$ and $\{\boldsymbol{\epsilon}_t\}_0^T \sim \mathcal{N}(0, \boldsymbol{I})$ denotes the standard Gaussian noises. Under the influence of this momentum field, for $\forall \, \boldsymbol{x}_0 \sim \boldsymbol{\pi}_0$ and $\boldsymbol{x}_1 \sim \boldsymbol{\pi}_1$, data $\boldsymbol{x}_0$ will gradually transform into noise $\boldsymbol{x}_1$ via the trajectory $T_{\boldsymbol{x}_0 \mapsto \boldsymbol{x}_1} = \{\boldsymbol{x}_0, \boldsymbol{z}_1, \cdots, \boldsymbol{z}_{T-1}, \boldsymbol{x}_1\}$. Note this momentum field $\{\boldsymbol{v}_t\}_0^{T-1}$ maintains the dynamics of the rectified flow (Liu et al., 2023a) during the initial noise-adding stage with the fastest initial vector $\boldsymbol{v}_0 = \beta(\boldsymbol{\epsilon}_0 - \boldsymbol{x}_0)$. As the velocity $\boldsymbol{v}_0$ is gradually noise-perturbed until approaching the noise $\beta\boldsymbol{\epsilon}_T$, we complete the progressively diverse modeling of an OT trajectory $T_{\boldsymbol{x}_0 \mapsto \boldsymbol{x}_1}$. Similar to DDPM (Ho et al., 2020), we can directly obtain the momentum $\boldsymbol{v}_t$ at any timestep $t$ via the one-step update formula as (see App. B for details),

$$\boldsymbol{v}_t = \sqrt{\bar{\gamma}_t}\boldsymbol{v}_0 + \sqrt{1 - \bar{\gamma}_t}\beta\boldsymbol{\epsilon}_t, \ \ \boldsymbol{v}_0 = \beta(\boldsymbol{\epsilon}_0 - \boldsymbol{x}_0),$$

(3)

where $\bar{\gamma}_t := \prod_{i=1}^{t} \gamma_i$. As derived from eq. (3), the proportion of $\boldsymbol{v}_0$ in $\boldsymbol{v}_t$ decays exponentially with increasing $t$. This indicates that during the forward process, the momentum $\boldsymbol{v}_t$ gradually deviates from the linear direction defined by $\boldsymbol{v}_0$, thereby progressively expanding the exploration diversity.

**Momentum Flow Matching Objective.** Building upon the flow matching framework for velocity field regression, we optimize the optimal transport (OT) problem by minimizing the MSE between

---

**Algorithm 1:** Momentum Flow Transport: Forward Process

1: **Procedure**: $T_o = \texttt{MomentumField}((\boldsymbol{z}_0, \boldsymbol{z}_T))$:
2: **Input:** $\boldsymbol{z}_0 \sim \boldsymbol{\pi}_0$, $\boldsymbol{z}_T = \boldsymbol{\epsilon}_0 \sim \boldsymbol{\pi}_1$, $T$, $\{\boldsymbol{\gamma}_t\}_1^{T-1}$, $\beta$, $\boldsymbol{v}_0 = \beta(\boldsymbol{\epsilon}_0 - \boldsymbol{z}_0)$.
3: **For** $t \leftarrow 1$ **to** $T$ **do repeat noise disturbance:**
 - $\boldsymbol{\epsilon}_t \sim \mathcal{N}(0, \boldsymbol{I})$.
 - $\boldsymbol{z}_t = \boldsymbol{z}_{t-1} + \boldsymbol{v}_{t-1}$.
 - $\boldsymbol{v}_t = \sqrt{\gamma_t}\boldsymbol{v}_{t-1} + \sqrt{1 - \gamma_t}\beta\boldsymbol{\epsilon}_t$.
4: **Return:** Trajectory $T_o = \{\boldsymbol{z}_0, \boldsymbol{z}_1, \cdots, \boldsymbol{z}_{T-1}, \boldsymbol{z}_T\}$.

---

predicted momentum and ground-truth. The momentum flow matching objective is formulated as:

$$\mathcal{L}_{\text{MFM}}(\theta) = \mathbb{E}_{t \sim U[0,1]}\|\boldsymbol{u}_\theta(\boldsymbol{z}_t, t) - \boldsymbol{v}_t\|^2, \tag{4}$$

where $\theta$ denotes learnable parameters for neural network $\boldsymbol{u}_\theta(\cdot, t)$, and $t \sim \mathcal{U}[0, 1]$. In the inference phase, once the momentum estimate $\boldsymbol{v}_{\theta;t} = \boldsymbol{u}_\theta(\cdot, t)$ is obtained, the reverse OT trajectory $\tilde{T}_\theta$ can be derived, as detailed in the subsequent forward and reverse processes.

### 3.2 Forward Process of the Momentum Flow

Let $\boldsymbol{z}_0 = \boldsymbol{x}_0 \sim \boldsymbol{\pi}_0$ and $\boldsymbol{z}_T = \boldsymbol{x}_1 \sim \boldsymbol{\pi}_1$ respectively denote the data and noise distributions on $\mathbb{R}^d$. When applying our momentum flow to the forward diffusion process, we can obtain the intermediate noisy distribution $\{\boldsymbol{z}_t \sim \boldsymbol{\pi}(\boldsymbol{z}_t)\}_1^{T-1}$ at discretized anchor points $\boldsymbol{z}_t$. In general, the forward diffusion process is defined as a Markov chain that progressively injects Gaussian noise $\boldsymbol{\epsilon}$ into $\boldsymbol{x}_0$ over $T$ timesteps according to the forward coefficient $a_t$ and $b_t$, which can be formally expressed as

$$q(\boldsymbol{z}_t|\boldsymbol{z}_{t-1}) = \mathcal{N}(\boldsymbol{z}_t; a_t\boldsymbol{z}_{t-1}, b_t^2\mathbf{I}), \; q(\boldsymbol{z}_{(1:T)}|\boldsymbol{z}_0) = \prod_{t=1}^{T} q(\boldsymbol{z}_t|\boldsymbol{z}_{t-1}). \tag{5}$$

We subsequently introduce our momentum field into the classical diffusion process to adjust the balance between optimal diversity and optimal efficiency for the exploration of the forward compromise trajectory $T_o$, and the detailed process is presented in Algorithm 1.

**Forward Momentum Flow.** From eq. (3), we can observe that the recursive formulation of our momentum field shares the similar form as that in DDPM (Ho et al., 2020), allowing us to directly obtain the prior probability distribution $q(\boldsymbol{v}_t|\boldsymbol{v}_{t-1})$ of the momentum flow:

$$q(\boldsymbol{v}_t|\boldsymbol{v}_{t-1}) = \mathcal{N}(\boldsymbol{v}_t; \sqrt{\alpha_t}\boldsymbol{v}_{t-1}, (1-\alpha_t)\beta^2\mathbf{I}), \; q(\boldsymbol{v}_t|\boldsymbol{v}_0) = \mathcal{N}(\boldsymbol{v}_t; \sqrt{\bar{\alpha}_t}\boldsymbol{v}_0, (1-\bar{\alpha}_t)\beta^2\mathbf{I}), \tag{6}$$

where $\alpha_t := \gamma$ and $\bar{\alpha}_t := \prod_{i=1}^{t} \alpha_i = \gamma^t$ ($\gamma < 1$ is a fixed constant). Notably, the formal alignment between the momentum flow and the forward process in DDPM (Ho et al., 2020) also allows a straightforward derivation of the posterior distribution $p_\theta(\boldsymbol{v}_{t-1}|\boldsymbol{v}_t)$ of momentum flow, see Sec. 3.3.

**Forward Data Flow.** Based on the above momentum flow, we can further build the forward trajectory (i.e., *data flow*) $T_o = \{\boldsymbol{z}_0, \boldsymbol{z}_1, \cdots, \boldsymbol{z}_{T-1}, \boldsymbol{z}_T\}$, which is represented in the form of a conditional probability distribution $q(\boldsymbol{z}_t|\boldsymbol{z}_{t-1})$. According to eq. (2) and eq. (3), the one-step forward data distribution can be obtained (see App. C for a detailed derivation) as

$$q(\boldsymbol{z}_t|\boldsymbol{z}_0) = \mathcal{N}\left(\boldsymbol{z}_t; (1 - (\frac{\sqrt{\gamma^t} - 1}{\sqrt{\gamma} - 1})\beta)\boldsymbol{z}_0, ((\frac{\sqrt{\gamma^t} - 1}{\sqrt{\gamma} - 1})^2 - \frac{\gamma^t - 1}{\gamma - 1} + t)\beta^2\mathbf{I}\right). \tag{7}$$

Due to $\beta := (\sqrt{\gamma} - 1)/(\sqrt{\gamma^T} - 1)$, when computing the noise distribution $\boldsymbol{\pi}(\boldsymbol{z}_T)$, we can eliminate the complex coefficient in front of $\boldsymbol{z}_0$ in eq. (7) to derive a *zero*-mean Gaussian distribution (independent of the data distribution $\boldsymbol{\pi}_0$), which can be formally expressed as

$$q(\boldsymbol{z}_T|\boldsymbol{z}_0) = \mathcal{N}\left(\boldsymbol{z}_T; 0, ((\frac{\sqrt{\gamma^T} - 1}{\sqrt{\gamma} - 1})^2 - \frac{\gamma^T - 1}{\gamma - 1} + T)\beta^2\mathbf{I}\right), \tag{8}$$

This simplified formula facilitates the subsequent reverse momentum transport process and significantly reduces computational complexity during training and inference.

---

**Algorithm 2:** Momentum Flow Matching: Reverse Process

1: **Procedure**: $\tilde{T}_\theta = \texttt{MomentumFlow}((\boldsymbol{z}_0, \boldsymbol{z}_T))$:
2: **Input:** Momentum model $\boldsymbol{u}_\theta(\cdot, t) : \mathbb{R}^d \times [0, 1] \to \mathbb{R}^d$ with parameters $\theta$.
3: **Training:** $\hat{\theta} = \arg\min_\theta \sum_{t=1}^T \mathbb{E}\left[\|\boldsymbol{u}_\theta(m\boldsymbol{z}_t + (1-m)\boldsymbol{z}_{t-1}, m) - (\boldsymbol{z}_t - \boldsymbol{z}_{t-1})\|^2\right]$,
   with $m \sim \mathcal{U}[0, 1]$.
4: **For** $t \leftarrow T$ **to** 1 **do repeat sampling:**
   - Draw $(\boldsymbol{z}_{t-1}, \boldsymbol{z}_t)$ from $\boldsymbol{\pi}(\boldsymbol{z}_{t-1}) \times \boldsymbol{\pi}(\boldsymbol{z}_t)$, with $\boldsymbol{z}_{t-1} \sim \boldsymbol{\pi}(\boldsymbol{z}_{t-1})$ and $\boldsymbol{z}_t \sim \boldsymbol{\pi}(\boldsymbol{z}_t)$.
   - Solve ODE: $\frac{d\boldsymbol{z}_t}{dt} = \boldsymbol{u}_\theta(\boldsymbol{z}_t^m, m)$, with $\boldsymbol{z}_0 \sim \boldsymbol{\pi}_0$.
   - Return: Sub-trajectory $\boldsymbol{z}_t = \{\boldsymbol{z}_t^m : m \in [0, 1]\}$.
5: **Return:** Trajectory $\tilde{T}_\theta = \{\boldsymbol{z}_t : t \in [0, 1]\}$.

---

### 3.3 Reverse Process of the Momentum Flow

The reverse process of the momentum flow aims to restore noise distribution $\boldsymbol{\pi}_1$ to data distribution $\boldsymbol{\pi}_0$ via an inverse trajectory $\tilde{T}_\theta = \{\boldsymbol{z}_T, \boldsymbol{z}_{T-1;\theta}, \cdots, \boldsymbol{z}_{1;\theta}, \boldsymbol{z}_{0;\theta}\}$, which is estimated by a neural network for approximating $\boldsymbol{z}_{t;\theta} \sim \boldsymbol{\pi}(\boldsymbol{z}_{t;\theta})$. To achieve this, we can approximate the momentum field $\{\boldsymbol{v}_t\}_0^{T-1}$ and then utilize the relationship $\boldsymbol{z}_{t-1;\theta} = \boldsymbol{z}_{t;\theta} - \boldsymbol{v}_{t-1;\theta}$ to estimate $\boldsymbol{z}_{t-1;\theta}$ from $\boldsymbol{z}_{t;\theta}$ and $\boldsymbol{v}_{t-1;\theta}$, as illustrated in Algorithm 2. We denote the estimated values of $(\boldsymbol{z}_t, \boldsymbol{v}_t)$ as $(\boldsymbol{z}_{t;\theta}, \boldsymbol{v}_{t;\theta})$. Based on this framework, we discuss two ways to approximate the momentum field. The first way is to approximate $\boldsymbol{v}_t$ by estimating $p_\theta(\boldsymbol{v}_{t-1}|\boldsymbol{v}_t)$. Benefiting from the formal similarity between the forward momentum flow and the DDPM formulation (Ho et al., 2020), we can derive directly the corresponding posterior distribution and estimate $\boldsymbol{v}_{t-1;\theta}$ from $\boldsymbol{v}_{t;\theta}$ by training a noise predictor $\boldsymbol{\epsilon}_\theta$:

$$p_\theta(\boldsymbol{v}_{t-1}|\boldsymbol{v}_t) = \mathcal{N}\left(\frac{\sqrt{\gamma}\left(1 - \gamma^{t-1}\right) + \sqrt{\gamma^{t-1}}(1-\gamma)}{1 - \gamma^t}\boldsymbol{v}_t - \frac{(1-\gamma)\beta\boldsymbol{\epsilon}_\theta}{\sqrt{\gamma(1-\gamma^t)}}, \frac{(1-\gamma)\left(1-\gamma^{t-1}\right)}{1-\gamma^t}\beta^2\mathbf{I}\right), \tag{9}$$

$$\boldsymbol{v}_{t-1;\theta} = \frac{\sqrt{\gamma}\left(1 - \gamma^{t-1}\right) + \sqrt{\gamma^{t-1}}(1-\gamma)}{1 - \gamma^t}\boldsymbol{v}_{t;\theta} - \frac{(1-\gamma)\beta\boldsymbol{\epsilon}_\theta}{\sqrt{\gamma(1-\gamma^t)}} + \sqrt{\frac{(1-\gamma)\left(1-\gamma^{t-1}\right)}{1-\gamma^t}}\beta\boldsymbol{\epsilon}. \tag{10}$$

The second method directly approximates $\boldsymbol{v}_t$ by employing rectified flow on each sub-path. Specifically, between each adjacent intermediate noise-perturbed distribution pair $(\boldsymbol{\pi}(\boldsymbol{z}_t), \boldsymbol{\pi}(\boldsymbol{z}_{t-1}))$ at the discretized anchor point pair $(\boldsymbol{z}_t, \boldsymbol{z}_{t-1})$, we insert $M$ intermediate points $\boldsymbol{z}_{t-1}^{(m)}$ via linear interpolation:

$$\boldsymbol{z}_{t-1}^{(m)} = m\boldsymbol{z}_t + (1-m)\boldsymbol{z}_{t-1}, \tag{11}$$

where $m \sim \mathcal{U}[0, 1]$. To enhance sampling efficiency, we apply rectified flow to formulate a straight path for each sub-path $\{\boldsymbol{\pi}(\boldsymbol{z}_t) \to \boldsymbol{\pi}(\boldsymbol{z}_{t-1})\}_1^T$, with the network $\boldsymbol{u}_\theta(\cdot, t)$ trained to match the corresponding velocity $\boldsymbol{v}_{t-1} = \boldsymbol{z}_t - \boldsymbol{z}_{t-1}$. Therefore, the original objective, i.e., eq. (4), is reformulated into the following optimization objective:

$$\mathcal{L}_{\text{MFM}}(\theta) = \sum_{t=1}^T \mathbb{E}_{t \sim \mathcal{U}[0,1]}\left[\|\boldsymbol{u}_\theta(m\boldsymbol{z}_t + (1-m)\boldsymbol{z}_{t-1}, m) - (\boldsymbol{z}_t - \boldsymbol{z}_{t-1})\|^2\right]. \tag{12}$$

The second method achieves much higher computational efficiency than DDPM by using rectified flow to optimize the trajectory. Therefore, we follow the second method in all experiments.

### 3.4 Momentum Flow Matching on SE(3)

We now describe the extension of our MFM to protein backbone generation. The backbone atom positions of each residue in a protein backbone are parameterized by a rigid transformation $T \in \text{SE}(3)$. Each frame $T = (r, x)$ consists of a rotation matrix $r \in \text{SO}(3)$ and a translation vector $x \in \mathbb{R}^3$. A protein backbone consists of $N$ residues meaning it can be parameterized as $\mathbf{T} = [T^{(1)}, \ldots, T^{(N)}]$ with $\mathbf{T} \in \text{SE}(3)^N$. For notational simplicity, our extension focuses on a single frame but applies to all frames in a backbone since $\text{SE}(3)^N$ is a product space and we use an additive metric over frames.

Different from previous methods where noise is directly superimposed on $T \in \text{SE}(3)$, we introduce noise on its tangent space $\mathfrak{se}(3)$ to characterize momentum. Specifically, the Lie algebra $\mathfrak{se}(3)$ consists of all infinitesimal generators of rigid body motions and can be formally represented as:

$$\mathfrak{se}(3) = \left\{ \begin{pmatrix} [\boldsymbol{\omega}]_\times & \boldsymbol{v} \\ 0 & 0 \end{pmatrix} \in \mathbb{R}^{4 \times 4} \ \middle| \ \boldsymbol{\omega}, \boldsymbol{v} \in \mathbb{R}^3 \right\}, \tag{13}$$

where $[\boldsymbol{\omega}]_\times$ denotes the rotation generator corresponding to the angular velocity $\boldsymbol{\omega}$ and $\boldsymbol{v}$ denotes the translation generator, i.e., linear velocity. Thus, each element in $\mathfrak{se}(3)$ is uniquely determined by 6 parameters $(\boldsymbol{\omega}, \boldsymbol{v}) \in \mathbb{R}^6$. Moreover, benefiting from the linear isomorphism $\mathfrak{se}(3) \cong \mathbb{R}^6$, calculations can be simplified smoothly from the complex nonlinear manifold $\text{SE}(3)$ to the vector space $\mathbb{R}^6$.

## 4 EXPERIMENTS

In this section, we conduct experiments using the rectified flow framework implemented in PyTorch to evaluate the image generation diversity and efficiency of the proposed momentum flow model. The primary objectives are to compare the generating performance between momentum flow and rectified flow, and to analyze the impact of the momentum field on the diversity and speed of the generative process. The results show that momentum flow retains the fast sampling capability of straight velocity fields. In addition, by injecting multi-scale noise through the momentum fields, the diversity and the quality of the generated images are significantly enhanced.

### 4.1 UNCONDITIONED IMAGE GENERATION

**Experiment Settings.** We build upon the official open-source implementation as the foundation of our model framework, and all experiments are conducted as illustrated in Table 4. To maximize performance within our computational budget, we conduct a grid search over learning rates and weight decay parameters. For evaluation, we generate $50,000$ samples from each model and evaluate generating quality and diversity using the Fréchet Inception Distance (FID) (Seitzer, 2020) and the recall value (Sajjadi et al., 2018). As 'recall' is defined as the coverage rate of generated samples over the real data distribution, we evaluate the diversity of generated samples by calculating the Recall value. Additional training and implementation details are provided in App. E.

**Comparison on CIFAR-10.** We report unconditional image generation results on the CIFAR-10 dataset (Krizhevsky, 2009). We train all models for 20,000 steps with a batch size of $1,024$. In Table 1, the FID-50K scores are obtained using 50-NFE sampling. All entries employ the same U-Net architecture, applied directly in the pixel space. On this dataset, our method demonstrates a clear advantage over prior approaches.

Table 1: Quantitative results on the CIFAR-10 dataset, while $\gamma = 0.99$ in our Momentum Flow.

| method | NFE | FID↓ |
|---|---|---|
| RectifiedFlow (Liu et al., 2022b) | 50 | 50.26 |
| NanoFlow (Zhu et al., 2024a) | 50 | 47.40 |
| MomentumFlow (ours) | 50 | **45.66** |

**Comparison on CelebA-HQ and ImageNet.** As shown in Table 2, the momentum flow model consistently achieves superior performance compared to the rectified flow model, as evidenced by significantly lower FID and higher recall values across various settings. The improvements on CelebA-HQ dataset are significant, achieving an average improvement of over 11 FID points and 0.06 recall values. In addition, when reducing the number of function evaluations (NFE) from 100 to 10, the performance degradation is minimal, and the model remains competitive with rectified flow under the same sampling budget, highlighting the efficiency of our method grounded in balanced transport (OT). These results indicate that the momentum flow model preserves the fast sampling efficiency of rectified flow while generating higher-quality images.

Table 2: Quantitative results on CelebA-HQ and ImageNet-64 datasets. Here $\widehat{Step}$ is the number of denoising steps in each sub-path ($\gamma = 0.98$).

| $N$ | $\widehat{Step}$ | CelebA-HQ | | | ImageNet-64 | | |
|---|---|---|---|---|---|---|---|
| | | FID ↓ | NFE ↓ | Recall ↑ | FID ↓ | NFE ↓ | Recall ↑ |
| 1 (**Rectified Flow**) | 10 | 98.98 | 10 | 0.268 | 61.48 | 10 | 0.366 |
| | 50 | 65.38 | 50 | 0.384 | 42.42 | 50 | 0.457 |
| | 100 | 58.90 | 100 | 0.445 | 41.83 | 100 | 0.451 |
| 2 (**Ours**) | 5 | 84.61 | 10 | 0.345 | 60.34 | 10 | 0.368 |
| | 25 | 54.07 | 50 | 0.457 | 41.99 | 50 | 0.454 |
| | 50 | **50.57** | 100 | **0.488** | **41.77** | 100 | **0.459** |
| 5 (**Ours**) | 2 | 110.72 | 10 | 0.177 | 104.06 | 10 | 0.258 |
| | 10 | 99.57 | 50 | 0.249 | 96.70 | 50 | 0.294 |
| | 20 | 94.61 | 100 | 0.261 | 94.87 | 100 | 0.294 |

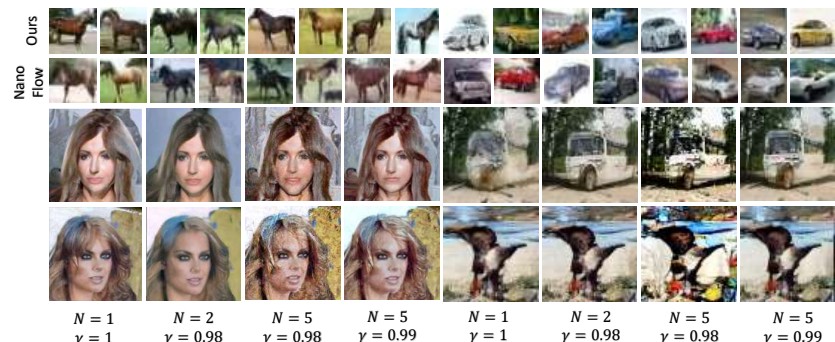

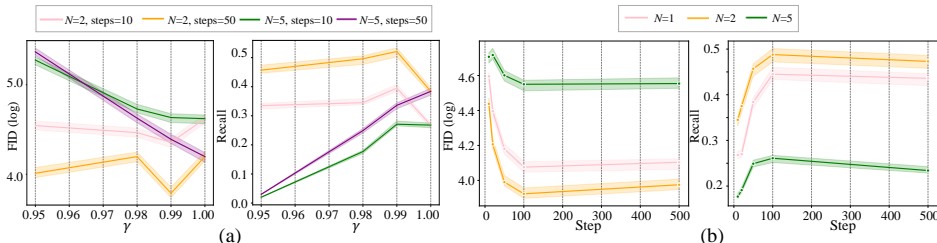

| $N = 1$ | $N = 2$ | $N = 5$ | $N = 5$ | $N = 1$ | $N = 2$ | $N = 5$ | $N = 5$ |
| $\gamma = 1$ | $\gamma = 0.98$ | $\gamma = 0.98$ | $\gamma = 0.99$ | $\gamma = 1$ | $\gamma = 0.98$ | $\gamma = 0.98$ | $\gamma = 0.99$ |

Figure 3: Samples of different datasets. Top: Samples on the CIFAR-10 dataset. Bottom: Momentum Flow samples with varying $N$ and $\gamma$, shown for CelebA-HQ (left) and ImageNet-64 (right).

Figure 4: (a) shows the relationship between model performance (FID, Recall) and the $\gamma$ setting. (b) illustrates the relationship between denoising steps and the quality of the generated images.

As shown in Figure 3, the deterministic nature of straight-line modeling in rectified flow leads to noticeable distortions in local details (e.g., mouth, eyes, and accessories). In contrast, the momentum flow model employs momentum-guided trajectories to explore a broader space, resulting in significantly improved detail generation. By dynamically injecting controllable velocity deviations via the momentum field, our method enhances both generating diversity and fidelity on high-resolution datasets such as CelebA-HQ, highlighting the effectiveness of multi-scale noise in guiding generation.

**Acceleration Process of Momentum Flow.** We empirically evaluate the efficiency of momentum flow in image generation. Although additional noise is injected into the velocity field, the linear straight structure ensures a constant velocity within each sub-path. This design preserves the efficiency of the original rectified flow. We compare the visual quality of generated images under different total denoising step settings: 10 and 50 steps while more examples can be found in the appendix . Momentum flow achieves comparable or even superior results to rectified flow with only half the number of sampling steps, as highlighted in the red-marked values in Table 2. These results demonstrate momentum flow inherits the acceleration advantages of rectified flows while further benefiting from enhanced flexibility.

**Sampling Efficiency and Generating Diversity.** Under a fixed image input, we compare the number of sampling steps under the same network architecture to assess their sampling efficiency. The Momentum Flow model achieves superior performance within the same time budget and requires fewer steps to reach comparable results, demonstrating its strong sampling efficiency. As shown in Figure 3, our method exhibits significant advantages in object color, shape, and background, and by adjusting $N$ and $\gamma$, Momentum Flow can produce more diverse colors and finer rendering details. These results confirm that our model effectively balances efficiency and diversity in the generative process, as visually shown in Figure 4.

**Momentum Decay Coefficient.** In our model framework, the decay coefficient $\gamma$ controls the level of noise perturbation by modulating the influence of the momentum flow, thereby enabling dynamic refinement of the forward trajectory. We observe that decreasing $\gamma$ causes the momentum flow to deviate more rapidly from the initial momentum direction $v_0$, which expands the exploration region and enhances sample diversity. However, excessive decay in the early stages may weaken the guidance from the initial momentum $v_0$, so $\gamma$ should not be set too low. As shown in Figure 4, when the number of noise-injecting steps is set to $N = 2$, a decay coefficient of $\gamma = 0.99$ results in significantly lower FID scores and higher recall values compared to $\gamma = 0.999$ and $\gamma = 0.98$. In contrast, when $N = 5$, the best performance is achieved with $\gamma = 1$, indicating that a larger number

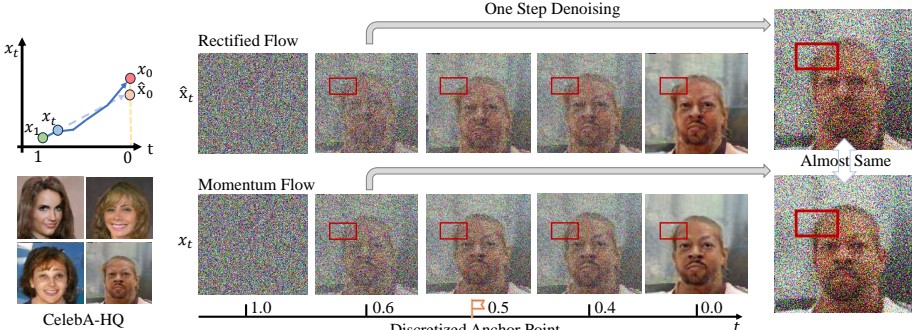

Figure 5: Reverse trajectories of momentum flow and rectified flow at different denoising steps, where the optimized velocity field of momentum flow improves image details.

of forward steps $N$ requires a slower decay (i.e., a $\gamma$ closer to 1) to maintain effective guidance of $v_0$. These experimental results suggest that appropriately selecting the decay coefficient $\gamma$ and the number of velocity steps $N$ can substantially improve both quality and diversity of generated images.

**Broader Analysis.** We compare the sampling processes of rectified flow and momentum flow to assess the advantages of our method in the denoising trajectory. In the early stages of generation—specifically the first few sampling steps—the results of both models appear similar. To illustrate this, we select $z_t$ at $t = 0.4$ and perform a single denoising step using the predicted velocity field to obtain a reference image, as shown on the right side of Figure 5. While the early-stage images generated by both methods show no notable differences, particularly in regions such as the hair, momentum flow exhibits a clear advantage in detail fidelity after passing the anchor point ($t = 0.5$).

Ignoring the refinement methods like distillation (Lee et al., 2024; Zhao et al., 2024), the initial velocity field often struggles to effectively bridge the gap between the noise and data distributions due to its reliance on fixed straight-line trajectories. The strength of our method lies in its ability to encourage broader exploration of the data space. By introducing momentum-guided velocity deviations, the model is not constrained to a fixed straight-line trajectory. Instead, it gains the flexibility to adjust its path dynamically.

## 4.2 RESULTS OF PROTEIN BACKBONE GENERATION

To verify the effectiveness of Momentum Flow on the protein monomer generation task, following GENIE (Lin & AlQuraishi, 2023) and FrameFlow (Yim et al., 2023a), we train it on the SCOPe dataset (Chandonia et al., 2022) with proteins below length 128 for a total of $3,938$

Table 3: Protein backbone generation results.

| Model | Sampling | Accuracy Metrics | | Confidence Metrics | |
|---|---|---|---|---|---|
| | | scTM ($> 0.5$)↑ | scRMSD↓ | pLDDT↑ | pAE↓ |
| GENIE | SDE | 0.09 (0.0) | 27.97 | 55.03 | 19.65 |
| FrameFlow | ODE | 0.39 (0.15) | 9.92 | 59.09 | 12.64 |
| **Ours** | MFM | **0.47 (0.45)** | **8.05** | **70.09** | **9.50** |

examples. During evaluation, we sample 10 backbones for every length between 60 and $300^1$ then use ProteinMPNN (Dauparas et al., 2022) to design 8 sequences for each backbone. We then evaluate the quality of generated proteins based on four metrics: scTM, scRMSD, pLDDT, and pAE. The quantitative results are reported in Table 3. See App. H for complete settings and detailed analysis.

## 5 CONCLUSIONS

Discretized-RF proposes a compromise transport method to balance the trade-off between diversity and efficiency. By injecting multi-scale noise perturbations based on momentum flow and formulating discretized straight-line trajectories, our approach effectively optimizes two key limitations of previous models: restricted generating diversity and high computational cost. Extensive experiments show that the momentum flow model achieves both high-quality image generation and fast sampling speed. Looking ahead, we believe the Discretized-RF framework offers a promising direction for designing more flexible flow trajectories and further exploring the diversity-efficiency optimal method.

---

[1] The upper limit of 300 here differs from the upper limit of 128 during training. We increase the upper limit during evaluation to demonstrate the generalization of our model in generating long sequence proteins.

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

## A ORGANIZATION OF THE SUPPLEMENTARY

In this supplementary, we first provide a detailed proof of the one-step momentum update in App. B. In App. C, we derive the one-step forward data distribution. In App. D, we conduct a toy experiment to illustrate the theoretical background of our momentum flow. In App. E, we present additional image experiments and more complete ablation studies. In particular, we recall in App. F some important theoretical preliminaries about SO(3) and SE(3). Using these, we introduce in App. G our protein backbone parameterization, the conversion between coordinates and frames, and the architecture of FramePred. In App. H, we show more detailed analysis of protein experiments.

## B PROOF OF THE ONE-STEP MOMENTUM UPDATE

In this section, we provide a detailed proof of the one-step update based on the recursive formulation (refer to eq. (2)) of the momentum field $\{\boldsymbol{v}_t\}_0^{T-1}$. The specific proof process is as follows:

$$
\begin{aligned}
\boldsymbol{v}_t &= \sqrt{\gamma_t}\boldsymbol{v}_{t-1} + \sqrt{1-\gamma_t}\beta\boldsymbol{\epsilon}_t \\
&= \sqrt{\gamma_t\gamma_{t-1}}\boldsymbol{v}_{t-2} + \sqrt{\gamma_t - \gamma_t\gamma_{t-1}}\beta\boldsymbol{\epsilon}_t + \sqrt{1-\gamma_t}\beta\boldsymbol{\epsilon}_{t-1} \\
&= \sqrt{\gamma_t\gamma_{t-1}}\boldsymbol{v}_{t-2} + \sqrt{1-\gamma_t\gamma_{t-1}}\beta\boldsymbol{\epsilon}_t \\
&= \cdots \\
&= \sqrt{\gamma_t\gamma_{t-1}\cdots\gamma_1}\boldsymbol{v}_0 + \sqrt{1-\gamma_t\gamma_{t-1}\cdots\gamma_1}\beta\boldsymbol{\epsilon}_t \\
&= \sqrt{\bar{\gamma}_t}\boldsymbol{v}_0 + \sqrt{1-\bar{\gamma}_t}\beta\boldsymbol{\epsilon}_t
\end{aligned}
\tag{14}
$$

where $\bar{\gamma}_t := \prod_{i=1}^{t}\gamma_i$.

## C PROOF OF THE ONE-STEP FORWARD DATA DISTRIBUTION

In the forward process, the data distribution evolves under the momentum field $\{\boldsymbol{v}_t\}_0^{T-1}$. According to the one-step momentum update formula (refer to eq. (3)), the forward data distribution $\boldsymbol{\pi}(\boldsymbol{z}_t)$ originates from the $\boldsymbol{\pi}_0$, perturbed by a exponentially scaled decaying contribution of the initial momentum $\boldsymbol{v}_0$, along with random noise $\boldsymbol{\epsilon_t}$. The specific derivation process is as follows:

$$
\begin{aligned}
\boldsymbol{z}_t &= \boldsymbol{z}_{t-1} + \boldsymbol{v}_{t-1} \\
&= \boldsymbol{z}_0 + \boldsymbol{v}_0 + \boldsymbol{v}_1 + \boldsymbol{v}_2 + \cdots + \boldsymbol{v}_{t-1} \\
&= \boldsymbol{z}_0 + \boldsymbol{v}_0(1 + \sqrt{\gamma} + \sqrt{\gamma^2} + \cdots + \sqrt{\gamma^{t-1}}) + \sum_{i=1}^{t-1}\sqrt{1-\gamma^i}\beta\boldsymbol{\epsilon}_i \\
&= \boldsymbol{z}_0 + \boldsymbol{v}_0(\frac{\sqrt{\gamma^t}-1}{\sqrt{\gamma}-1}) + \sqrt{t - (1+\gamma+\gamma^2+\cdots+\gamma^{t-1})}\beta\boldsymbol{\epsilon}_t \\
&= \boldsymbol{z}_0 + (\boldsymbol{\epsilon}_0 - \boldsymbol{z}_0)(\frac{\sqrt{\gamma^t}-1}{\sqrt{\gamma}-1})\beta + \sqrt{t - \frac{\gamma^t-1}{\gamma-1}}\beta\boldsymbol{\epsilon}_t \\
&= (1 - (\frac{\sqrt{\gamma^t}-1}{\sqrt{\gamma}-1})\beta)\boldsymbol{z}_0 + \sqrt{(\frac{\sqrt{\gamma^t}-1}{\sqrt{\gamma}-1})^2 - \frac{\gamma^t-1}{\gamma-1} + t}\beta\boldsymbol{\epsilon}_t.
\end{aligned}
\tag{15}
$$

Thus we have

$$
q(\boldsymbol{z}_t|\boldsymbol{z}_0) = \mathcal{N}\left(\boldsymbol{z}_t; (1 - (\frac{\sqrt{\gamma^t}-1}{\sqrt{\gamma}-1})\beta)\boldsymbol{z}_0, ((\frac{\sqrt{\gamma^t}-1}{\sqrt{\gamma}-1})^2 - \frac{\gamma^t-1}{\gamma-1} + t)\beta^2\mathbf{I}\right).
\tag{16}
$$

## D MFM TOY EXPERIMENT

### D.1 TOY EXPERIMENT PARAMETERIZATION

To illustrate the theoretical background of our momentum flow, we provide an example in Figure 6, demonstrating the expected momentum flow in the forward process and the optimal transport path in

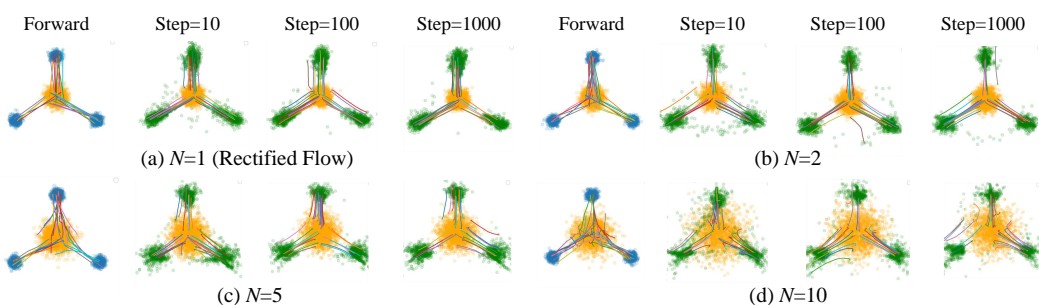

Forward      Step=10      Step=100      Step=1000      Forward      Step=10      Step=100      Step=1000

(a) *N*=1 (Rectified Flow)                                              (b) *N*=2

(c) *N*=5                                              (d) *N*=10

Figure 6: Forward and reverse trajectories of our momentum flow with different numbers $N$ of discretized anchor points. In our settings, blue points are sampled from $\pi_0$, orange points are sampled from $\pi_1$, green points denote generated samples, and lines represent transport trajectories.

the reverse process. Momentum flow is simulated utilizing the Euler method with a constant step size of $1/N$, computed at $N$ discrete anchor points, where $N$ denotes the number of such points, corresponding to the value of $T$ in Momentum Flow. The number of sampling points on each segment is defined as $\widehat{Step}$, and the total number of sampling steps is defined as $Step = \widehat{Step} \times N$. Note that in all experiments in this section, this notation is used by default. Moreover, we define the use of a fully connected neural network with two hidden layers to estimate the momentum field. In practice, the model is trained using full-batch gradient descent and optimized with the Adam optimizer.

## D.2 ADDITIONAL ANALYSIS FOR MFM TOY EXPERIMENTS

As shown in Figure 6, increasing the number of discretized anchor points causes significant fluctuations in the velocity field near $\pi_0$ during the forward process, highlighting the impact of trajectory complexity on learning. Furthermore, our method encourages exploration of diverse trajectories, as evidenced by the "turning-back" phenomenon observed in the early stages of the reverse process when the number of discretized anchor points ($N$) increases. This allows for more exploration in the space rather than directly pointing to the $\pi_0$ distribution. Despite the unpredictability of the varying velocity field, the residual correlation between the forward and reverse velocity fields, enabled by the momentum field, facilitates velocity field prediction. Additionally, the piecewise linear nature of the trajectory preserves the accelerated denoising capability of Rectified Flow, enabling the generation of high-quality samples with a small number of steps while maintaining high denoising efficiency.

## E ADDITIONAL IMAGE EXPERIMENTS

### E.1 EXPERIMENT DETAILS

**Dataset Description:** We use three datasets for training, including

1. CIFAR-10: Images with a resolution of $32 \times 32$ from the CIFAR-10 training split.

2. CelebA-HQ: Images from the 'img_align_celeba_png.7z' version of the CelebA-HQ dataset, resized to $256 \times 256$ resolution.

3. ImageNet: Images from ImageNet resized respectively to $32 \times 32$ and $64 \times 64$ resolutions.

Note that during training, images are normalized to have zero mean and unit variance.

**Implementation Details:** The experiments are implemented in PyTorch (version 2.6.0) and conducted on an NVIDIA A800-SXM4-80GB GPU. Random seeds are set to $42$ for reproducibility.

**Performance Details:** Given that our training commenced from scratch and employed a relatively simple network architecture (U-net), our baseline performance is not as robust as that of most diffusion models with more intricate designs. However, since all our experiments were conducted within the same architectural framework that we designed, our comparisons remain fair and persuasive.

**Additional Notes:** Ablation studies are conducted to analyze the impact of different hyperparameters. Hyperparameter tuning is performed using grid search over the learning rate and batch size.

| Dataset | CIFAR-10 | CelebA-HQ | ImageNet | Scope |
|---|---|---|---|---|
| resolution | 32 | 256 | 64 | - |
| params (M) | 35 | 120 | 120 | 45 |
| step | 20k | 70k | 70k | 1k |
| batch size | 1024 | 128 | 512 | 40 |
| optimizer | | Adam | | |
| learning rate | | $3e-4$ | | |
| ema decay | | 0.9999 | | |

Table 4: Configurations for different datasets.

### E.2 EXPERIMENTS ON CELEBA-HQ AND IMAGENET

As shown in Table 2, the momentum flow model consistently achieves significant improvements on CelebA-HQ and ImageNet. By balancing $N$ and $\gamma$,we verify that more anchor points bring greater gains and $\gamma$ should increase with the number of anchor points to reduce velocity abruptness in momentum field.As shown in 7 and 8xuezh, a larger $N$ brings more significant diversity, but it may compromise image quality, so the noise intensity $\gamma$ should be correspondingly adjusted. The momentum flow model retains the fast sampling efficiency of rectified flow, while also generating higher-quality images, as shown by these results.

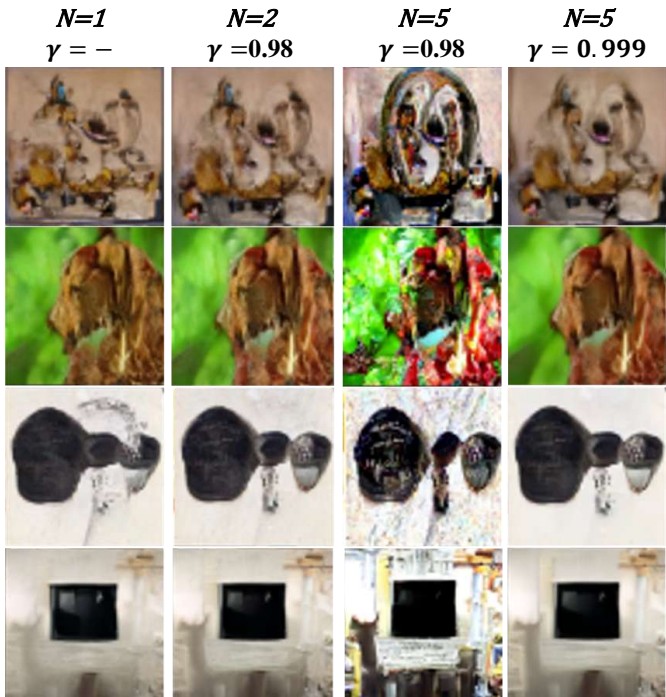

Figure 7: The impact of adjusting $N$ and $\gamma$ on image details in Momentum Flow ($Step = 50$). Here '–' means that $\gamma$ do not influence the trajectories while $N = 1$.

$\gamma = 0.95$  $\gamma = 0.98$  $\gamma = 0.99$  $\gamma = 0.999$  $\gamma = 1.0$

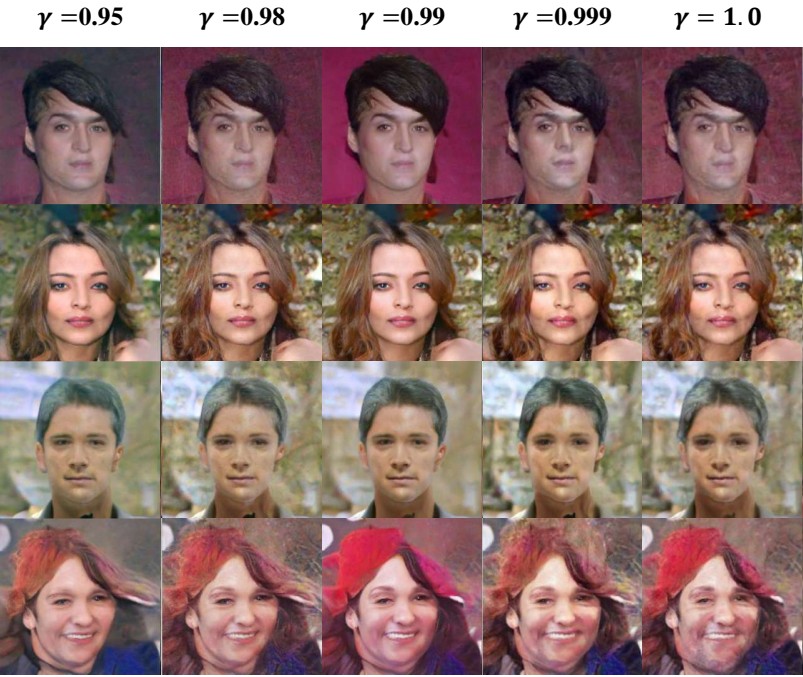

Figure 8: Generated images under different values of gamma, where both excessively large and excessively small gamma values can deteriorate image quality ($N = 2, Step = 50$).

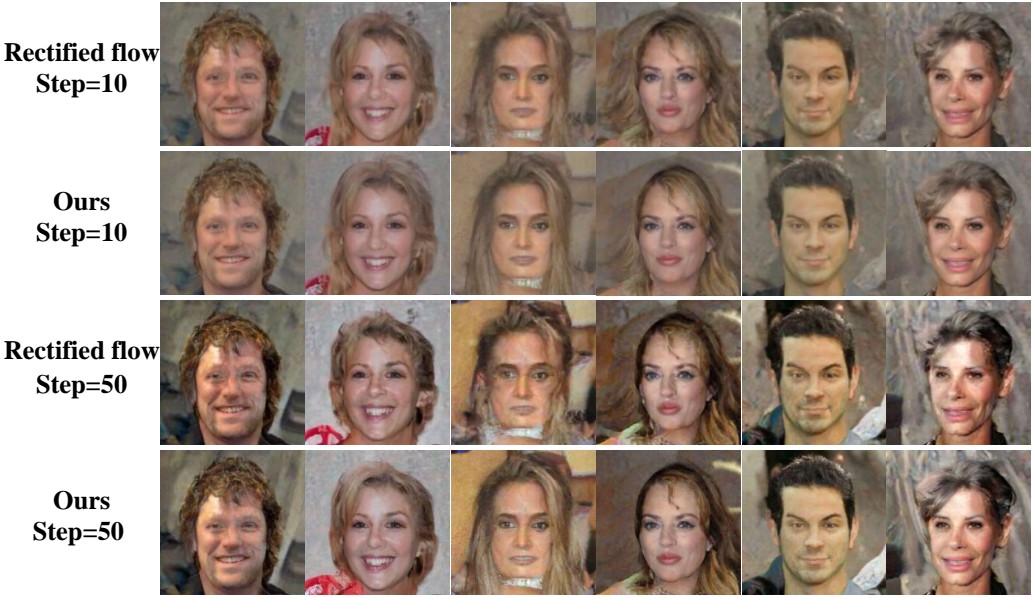

Figure 9: Face generation by Rectified Flow and Momentum Flow under different sampling steps.

## F    THEORETICAL PRELIMINARIES ABOUT SO(3) AND SE(3)

This section synthesizes the theoretical foundations of Lie groups SO(3) and SE(3) from two complementary perspectives: geometric structure and representation theory. By integrating these theories, we establish a rigorous mathematical toolkit for rigid-body transformations in computational biology.

### F.1 SO(3) LIE GROUP

The Special Orthogonal group in 3 dimensions, SO(3) consists of the 3D rotation matrices:

$$\text{SO}(3) = \left\{ r \in \mathbb{R}^{3 \times 3} : r^\top r = r r^\top = I, \det r = 1 \right\}. \tag{17}$$

#### F.1.1 LIE ALGEBRA OF SO(3) AND HAT OPERATION

SO(3) is a matrix Lie group and its Lie algebra $\mathfrak{so}(3)$ consists of all $3 \times 3$ skew-symmetric matrices:

$$\mathfrak{so}(3) = \left\{ \mathfrak{r} \in \mathbb{R}^{3 \times 3} : \mathfrak{r}^\top = -\mathfrak{r} \right\}. \tag{18}$$

$\mathfrak{so}(3)$ is 3-dimensional and is isomorphic to the $\mathbb{R}^3$ vector via the *hat* operation $\widehat{(\cdot)} : \mathbb{R}^3 \to \mathfrak{so}(3)$ as

$$\mathfrak{r} = \widehat{\boldsymbol{\omega}} = \begin{pmatrix} 0 & -\omega_3 & \omega_2 \\ \omega_3 & 0 & -\omega_1 \\ -\omega_2 & \omega_1 & 0 \end{pmatrix} \in \mathfrak{so}(3), \ \forall \boldsymbol{\omega} = (\omega_1, \omega_2, \omega_3)^\top \in \mathbb{R}^3. \tag{19}$$

The matrix $\mathfrak{r}$ can be uniquely identified with a vector $\boldsymbol{\omega} \in \mathbb{R}^3$ such that $\forall \mathbf{v} \in \mathbb{R}^3$, $\mathfrak{r}\mathbf{v} = \widehat{\boldsymbol{\omega}}\mathbf{v} = \boldsymbol{\omega} \times \mathbf{v}$, where $\times$ indicates the cross product. The vector $\boldsymbol{\omega}$ is known as the *rotation vector*, i.e., *angular velocity*. Moreover, the Lie bracket on $\mathfrak{so}(3)$ corresponds to the cross product in $\mathbb{R}^3$:

$$[\widehat{\boldsymbol{\omega}}_1, \widehat{\boldsymbol{\omega}}_2] = \widehat{\boldsymbol{\omega}}_1 \widehat{\boldsymbol{\omega}}_2 - \widehat{\boldsymbol{\omega}}_2 \widehat{\boldsymbol{\omega}}_1 = \widehat{\boldsymbol{\omega}}_1 \times \boldsymbol{\omega}_2. \tag{20}$$

This $\mathfrak{so}(3)$-$\mathbb{R}^3$ isomorphism allows the rotation vector $\boldsymbol{\omega} \in \mathbb{R}^3$ to encode both rotation axis (direction of $\boldsymbol{\omega}$) and angle ($\|\boldsymbol{\omega}\|$) in a unified framework. Specifically, the magnitude of this vector, $\theta = \|\boldsymbol{\omega}\|$ denotes the *angle* of rotation and the direction of this vector, $e_{\boldsymbol{\omega}} = \frac{\boldsymbol{\omega}}{\|\boldsymbol{\omega}\|}$ denotes the *axis* of rotation.

#### F.1.2 PARAMETERIZATIONS OF SO(3)

Here we describe two different possible parameterizations of SO(3) and its Lie algebra $\mathfrak{so}(3)$.

**Axis-angle.** Let a unit vector $e_{\boldsymbol{\omega}} = (a, b, c) \in \mathbb{S}^2$ represent the rotation axis, where $(a, b, c) \in \mathbb{R}^3$ and $a^2 + b^2 + c^2 = 1$, and $\theta \in \mathbb{R}_+$ represent the rotation angle. Hence, any rotation matrix in SO(3) can be formally written via the exponential mapping as $r = \exp(\mathfrak{r}) \in \text{SO}(3)$, where $\mathfrak{r} = \theta X \in \mathfrak{so}(3)$ and $X = aX_1 + bX_2 + cX_3{}^2$. The parameterization of SO(3) using $(e_{\boldsymbol{\omega}}, \theta)$ is called the *axis-angle* theory. Notably, $X^3 = -X$ and the explicit form of $r$ can be given by the *Rodrigues' formula* as

$$r = \exp(\theta X) = I + \sin\theta \cdot X + (1 - \cos\theta)X^2, \tag{21}$$

which provides a concise way of computing the exponential. In addition, for $\forall (a, b, c), \mathbf{v} \in \mathbb{R}^3$,

$$X\mathbf{v} = (aX_1 + bX_2 + cX_3)\mathbf{v} = e_{\boldsymbol{\omega}} \times \mathbf{v}, \ X^2\mathbf{v} = (aX_1 + bX_2 + cX_3)^2\mathbf{v} = \langle e_{\boldsymbol{\omega}}, \mathbf{v} \rangle e_{\boldsymbol{\omega}} - \mathbf{v}. \tag{22}$$

Then substitute these into the expression above to obtain the *Rodrigues' rotation* formula:

$$r\mathbf{v} = \exp(\theta X)\mathbf{v} = \cos\theta \cdot \mathbf{v} + \sin\theta \cdot (e_{\boldsymbol{\omega}} \times \mathbf{v}) + (1 - \cos\theta)\langle e_{\boldsymbol{\omega}}, \mathbf{v} \rangle e_{\boldsymbol{\omega}}. \tag{23}$$

As this formula shows, $\exp(\theta X)\mathbf{v}$ denotes the rotation of the vector $\mathbf{v}$ of angle $\theta$ around the axis $e_{\boldsymbol{\omega}}$. Moreover, an equivalent representation defines the rotation matrix as $r = \exp(\mathfrak{r}) = \exp(\widehat{\boldsymbol{\omega}})$, where $\widehat{\boldsymbol{\omega}} \in \mathfrak{so}(3)$ and $\boldsymbol{\omega} = \|\boldsymbol{\omega}\|\frac{\boldsymbol{\omega}}{\|\boldsymbol{\omega}\|} = \theta\, e_{\boldsymbol{\omega}}$ is the rotation vector. So there exists another expression of $r$:

$$r = \exp(\widehat{\boldsymbol{\omega}}) = I + \frac{\sin\theta}{\theta}\widehat{\boldsymbol{\omega}} + \frac{1 - \cos\theta}{\theta^2}\widehat{\boldsymbol{\omega}}^2. \tag{24}$$

Notably, it is continuous at $\theta = 0$, yielding the identity matrix $I$. And the vector rotation formula is:

$$r\mathbf{v} = \exp(\widehat{\boldsymbol{\omega}})\mathbf{v} = \cos\theta \cdot \mathbf{v} + \frac{\sin\theta}{\theta}(\boldsymbol{\omega} \times \mathbf{v}) + \frac{1 - \cos\theta}{\theta^2}\langle \boldsymbol{\omega}, \mathbf{v} \rangle \boldsymbol{\omega}. \tag{25}$$

**Euler angles.** Rotation can also be decomposed into sequential elementary rotations about coordinate axes. A common convention is to utilize the $x$-convention with three angles $(\psi, \phi, \varphi) \in \mathbb{R}^3$. Specifically, the rotation is given by: a rotation about the $z$-axis by $\psi$, a second rotation about the former $x$-axis by $\phi$, and a last one about the former $z$-axis by $\varphi$. It can be formally expressed as

$$r = \exp[\psi X_3] \exp[\phi X_1] \exp[\varphi X_3]$$
$$= \begin{pmatrix} \cos(\psi) & -\sin(\psi) & 0 \\ \sin(\psi) & \cos(\psi) & 0 \\ 0 & 0 & 1 \end{pmatrix} \begin{pmatrix} 1 & 0 & 0 \\ 0 & \cos(\phi) & -\sin(\phi) \\ 0 & \sin(\phi) & \cos(\phi) \end{pmatrix} \begin{pmatrix} \cos(\varphi) & -\sin(\varphi) & 0 \\ \sin(\varphi) & \cos(\varphi) & 0 \\ 0 & 0 & 1 \end{pmatrix}. \tag{26}$$

Technically speaking, these three angles $\psi, \phi, \varphi$ are called the *Euler angles*: $\psi$ is called the *precession angle*, $\phi$ is called the *nutation angle*, and $\varphi$ is called the *angle of proper rotation* (or *spin*).

---

$^2 X$ is a skew-symmetric matrix in $\mathfrak{so}(3)$ and $(X_1, X_2, X_3)$ is the standard basis of $\mathfrak{so}(3)$.

### F.1.3 METRIC ON SO(3)

The metric on a Lie group $G$ is a smooth assignment of the inner product to each of its tangent space $\mathcal{T}_g G^3$, where $g \in G$. Thus, a common way to construct a metric on $G$ is to first define the inner product on $\mathfrak{g}$ and then extend it to the entire group via left (or right) translation. A particularly important class is the *bi-invariant* metric, which maintains invariant under both left and right translations.

Let $Q$ be a symmetric positive-definite matrix defining a quadratic form on $\mathfrak{g}$, formally expressed as

$$Q = \begin{pmatrix} A & B^\top \\ B & C \end{pmatrix}. \tag{27}$$

Depending on the matrix $Q$, the inner product between two elements $X_0, Y_0 \in \mathfrak{g}$ is given by

$$\langle X_0, Y_0 \rangle_{\mathfrak{g}} = \text{tr}(X_0^\top Q Y_0). \tag{28}$$

Then this inner product on $\mathfrak{g}$ can be extended to a left-invariant metric on $g \in G$ via

$$\langle u, v \rangle_{\mathcal{T}_g G} = \langle dL_{g^{-1}} u, dL_{g^{-1}} v \rangle_{\mathfrak{g}}, \ \forall u, v \in \mathcal{T}_g G. \tag{29}$$

Specifically, to calculate the inner product of two tangent vectors $u, v \in \mathcal{T}_g G$ at $g \in G$, we first use the differential $dL_{g^{-1}}(\cdot)$[4] to "pull back" $u, v$ to the identity element $\mathbf{e}$, thus becoming $U = dL_{g^{-1}}(u) \in \mathfrak{g}$ and $V = dL_{g^{-1}}(v) \in \mathfrak{g}$, and then use the inner product $\langle \cdot, \cdot \rangle_{\mathfrak{g}}$ defined in advance to calculate the inner product of $U$ and $V$. This result is further defined as the inner product of $u$ and $v$ at $g \in G$. Similarly, this inner product on $\mathfrak{g}$ can also be extended to a right-invariant metric on $h \in G$ via

$$\langle m, n \rangle_{\mathcal{T}_h G} = \langle dR_{h^{-1}} m, dR_{h^{-1}} n \rangle_{\mathfrak{g}}, \ \forall m, n \in \mathcal{T}_h G. \tag{30}$$

To sum up, the metric is bi-invariant. Moreover, a canonical choice for the metric of SO(3) is obtained by taking $Q = 1/2I$, resulting in a bi-invariant metric on SO(3). Therefore, the metric is given by

$$\langle \mathfrak{r}_1, \mathfrak{r}_2 \rangle_{\mathfrak{so}(3)} = \text{tr}(\mathfrak{r}_1^\top Q \mathfrak{r}_2) = \frac{1}{2} \text{tr}(\mathfrak{r}_1^\top \mathfrak{r}_2), \ \forall \mathfrak{r}_1, \mathfrak{r}_2 \in \mathfrak{so}(3). \tag{31}$$

Note that the inner product on *Lie groups* essentially acts on elements of the *Lie algebra* and, since the left action is transitive, this inner product is well-defined for all tangent spaces of the group elements. To further verify the bi-invariance of the SO(3) metric, consider the adjoint action $\text{Ad}_r(\mathfrak{r}) = r\mathfrak{r}r^\top$ for $\forall r \in \text{SO}(3)$ and $\mathfrak{r} \in \mathfrak{so}(3)$. Then the specific action process can be formally expressed as

$$\langle \text{Ad}_r \mathfrak{r}_1, \text{Ad}_r \mathfrak{r}_2 \rangle_{\mathfrak{so}(3)} = \frac{1}{2} \text{tr}\left((r\mathfrak{r}_1 r^\top)^\top (r\mathfrak{r}_2 r^\top)\right) = \frac{1}{2} \text{tr}(r\mathfrak{r}_1^\top \mathfrak{r}_2 r^\top) = \frac{1}{2} \text{tr}(\mathfrak{r}_1^\top \mathfrak{r}_2) = \langle \mathfrak{r}_1, \mathfrak{r}_2 \rangle_{\mathfrak{so}(3)}, \tag{32}$$

where we utilize the cyclic property of the trace[5] and the orthogonality of $r$, i.e., $r^\top r = I$. This result shows the metric is invariant under the adjoint action, which implies bi-invariance on SO(3).

The geodesic distance between two elements $r_1, r_2 \in \text{SO}(3)$ induced by this metric is given by

$$d_{\text{SO}(3)}(r_1, r_2) = ||\log(r_1^\top r_2)||_F, \tag{33}$$

where $\log$ is the matrix logarithm mapping and $|| \cdot ||_F$ is the Frobenius norm.

### F.1.4 THE ISOTROPIC GAUSSIAN DISTRIBUTION ON SO(3)

$\mathcal{IG}_{\mathbf{SO}(3)}$ **density.** The isotropic Gaussian distribution on SO(3), denoted as $\mathcal{IG}_{\text{SO}(3)}$, is parameterized by a mean rotation $r \in \text{SO}(3)$ and a concentration parameter $\epsilon \in \mathbb{R}$. It can be expressed in the *axis–angle* representation (refer to App. F.1.2), where the rotation axis is sampled uniformly and the rotation angle $\theta$ follows a probability density function (abbreviated as pdf) given by

$$f(\theta, \epsilon) = \sum_{l=0}^{\infty} (2l+1) e^{-l(l+1)\epsilon} \frac{\sin((l+1/2)\theta)}{\sin(\theta/2)}. \tag{34}$$

Although this expression involves a complex infinite series, Matthies et al. (1988) has shown that for $\epsilon \leqslant 1$, it can be accurately approximated by a closed-form expression:

$$f(\theta, \epsilon) = \sqrt{\pi} \epsilon^{-3/2} e^{\frac{\epsilon - \theta^2/\epsilon}{4}} \frac{\left(\theta - e^{-\pi^2/\epsilon} \left((\theta - 2\pi) e^{\pi\theta/\epsilon} + (\theta + 2\pi) e^{-\pi\theta/\epsilon}\right)\right)}{2 \sin\left(\frac{\theta}{2}\right)}. \tag{35}$$

---

[3]At the identity element $\mathbf{e} \in G$, the tangent space $\mathcal{T}_{\mathbf{e}} G$ coincides with the Lie algebra $\mathfrak{g}$.

[4]The notation $dL_{g^{-1}}(\cdot)$ is standard in differential geometry, where $L_{g^{-1}}$ denotes left translation by $g^{-1}$, defined as $L_{g^{-1}}(h) = g^{-1}h$. And the differential $dL_{g^{-1}}$ is a linear mapping that pull back a tangent vector $u \in \mathcal{T}_g G$ at $g \in G$ to the tangent space at the identity element $\mathbf{e}$, i.e., $dL_{g^{-1}} : \mathcal{T}_g G \to \mathcal{T}_{\mathbf{e}} G = \mathfrak{g}$.

[5]This property is embodied in: $\frac{1}{2} \text{tr}(r\mathfrak{r}_1^\top \mathfrak{r}_2 r^\top) = \frac{1}{2} \text{tr}(r(\mathfrak{r}_1^\top \mathfrak{r}_2)r^\top) = \frac{1}{2} \text{tr}((\mathfrak{r}_1^\top \mathfrak{r}_2)r^\top r) = \frac{1}{2} \text{tr}(\mathfrak{r}_1^\top \mathfrak{r}_2)$.

**Sampling from $\mathcal{IG}_{\mathbf{SO(3)}}$.** Sampling from $\mathcal{IG}_{\mathrm{SO}(3)}$ follows the procedure described by Leach et al. (2022). The rotation angle $\theta$ is obtained via inverse transform sampling, using the cumulative distribution function (abbreviated as cdf) derived from the pdf above. And this cdf is normalized appropriately, accounting for the uniform base density on SO(3), i.e., $f(\theta) = \frac{1-\cos\theta}{\pi}$. Moreover, the rotation axis is sampled uniformly from the two-sphere $\mathbb{S}^2$. Notably, The closed-form approximation of eq. (35) achieves fast computation of the cdf, thus making the sampling process highly efficient.

### F.2 SE(3) LIE GROUP

The Special Euclidean group, denoted as SE(3), constitutes the set of all rigid-body transformations (including rotation and translation) in three-dimensional space. It can be formally defined as

$$\mathrm{SE}(3) = \left\{ \begin{pmatrix} r & \boldsymbol{v} \\ 0 & 1 \end{pmatrix} : r \in \mathrm{SO}(3), \boldsymbol{v} \in (\mathbb{R}^3, +) \right\}, \tag{36}$$

where each element is represented by a $4 \times 4$ matrix. And endowed with the group operation of matrix multiplication, SE(3) can also be seen as a subgroup of the general linear group $\mathrm{GL}(4, \mathbb{R})$.

The corresponding Lie algebra of the Lie group SE(3), i.e., $\mathfrak{se}(3)$, is given by

$$\mathfrak{se}(3) = \left\{ \xi = \begin{pmatrix} \mathfrak{r} & \boldsymbol{v} \\ 0 & 0 \end{pmatrix} : \mathfrak{r} \in \mathfrak{so}(3), \boldsymbol{v} \in \mathbb{R}^3 \right\}, \tag{37}$$

where $\mathfrak{r}$ can also be denoted as $[\boldsymbol{\omega}]_\times$, indicating the skew-symmetric matrix form of its axis-angle representation $\boldsymbol{\omega} \in \mathbb{R}^3$. Note that the tangent space of the translation group $(\mathbb{R}^3, +)$ is isomorphic to $\mathbb{R}^3$ itself so we can directly use the notation $\boldsymbol{v}$ instead of $\mathfrak{v}$. Hence, each element $\xi \in \mathfrak{se}(3)$ is uniquely determined by 6 parameters $(\boldsymbol{\omega}, \boldsymbol{v}) \in \mathbb{R}^6$ and there further exists an isomorphism between $\mathfrak{se}(3)$ and $\mathbb{R}^6$ via the mapping: $\xi \mapsto (\boldsymbol{\omega}, \boldsymbol{v})$[6]. Moreover, since the translation group $(\mathbb{R}^3, +)$ is a normal subgroup of SE(3), the full group can be written as a semi-direct product: $\mathrm{SE}(3) = \mathrm{SO}(3) \ltimes (\mathbb{R}^3, +)$.

**Metric on SE(3).** While numerous metrics can be defined on SE(3), none of them are bi-invariant. Thus, it is common to construct either a left- or right-invariant metric. A straightforward choice for the quadratic form $Q$ from eq. (27) is setting the matrices $A = C = I_3$ and $B = 0$ (Park & Brockett, 1994). Consequently, the matrix $Q$ after the assignment can be formally expressed as

$$Q = \begin{pmatrix} I_3 & 0 \\ 0 & I_3 \end{pmatrix}. \tag{38}$$

Utilizing this metric we can define an inner product on SE(3) as $\langle \xi_1, \xi_2 \rangle_{\mathfrak{se}(3)} = \mathrm{tr}(\xi_1^\top Q \xi_2)$[7], where $\mathrm{tr}$ is the trace operation. For $\xi_1, \xi_2 \in \mathfrak{se}(3)$, the inner product expands explicitly as

$$\mathrm{tr}(\xi_1^\top Q \xi_2) = \mathrm{tr}\left( \begin{pmatrix} \mathfrak{r}_1 & \boldsymbol{v}_1 \\ 0 & 0 \end{pmatrix}^\top \begin{pmatrix} I_3 & 0 \\ 0 & I_3 \end{pmatrix} \begin{pmatrix} \mathfrak{r}_2 & \boldsymbol{v}_2 \\ 0 & 0 \end{pmatrix} \right) = \mathrm{tr}\begin{pmatrix} \mathfrak{r}_1^\top \mathfrak{r}_2 & \mathfrak{r}_1^\top \boldsymbol{v}_2 \\ \boldsymbol{v}_1^\top \mathfrak{r}_2 & \boldsymbol{v}_1^\top \boldsymbol{v}_2 \end{pmatrix}. \tag{39}$$

After further derivation, we can obtain: $\mathrm{tr}(\xi_1^\top Q \xi_2) = \mathrm{tr}(\mathfrak{r}_1^\top \mathfrak{r}_2) + \mathrm{tr}(\boldsymbol{v}_1^\top \boldsymbol{v}_2) = \mathrm{tr}(\mathfrak{r}_1^\top Q \mathfrak{r}_2) + \boldsymbol{v}_1^\top \boldsymbol{v}_2$[8]. Therefore, the metric on SE(3) can be formally decomposed into the metrics on SO(3) and $\mathbb{R}^3$:

$$\langle \xi_1, \xi_2 \rangle_{\mathfrak{se}(3)} = \langle \mathfrak{r}_1, \mathfrak{r}_2 \rangle_{\mathfrak{so}(3)} + \langle \boldsymbol{v}_1, \boldsymbol{v}_2 \rangle_{\mathbb{R}^3}. \tag{40}$$

Hence, geodesics on SE(3) can be derived from those on the product manifold $\mathrm{SO}(3) \times \mathbb{R}^3$ and the distance between $x_1 = (r_1, \boldsymbol{v}_1) \in \mathrm{SE}(3)$ and $x_2 = (r_2, \boldsymbol{v}_2) \in \mathrm{SE}(3)$ is given by

$$d_{\mathrm{SE}(3)}(x_1, x_2) = \sqrt{d_{\mathrm{SO}(3)}(r_1, r_2)^2 + d_{\mathbb{R}^3}(\boldsymbol{v}_1, \boldsymbol{v}_2)^2}. \tag{41}$$

where $d_{\mathrm{SO}(3)}$ is defined in eq. (33) and $d_{\mathbb{R}^3}$ denotes the standard Euclidean distance.

---

[6]The isomorphism between $\mathfrak{se}(3)$ and $\mathbb{R}^6$ identifies each element $\xi \in \mathfrak{se}(3)$ with a twist comprising rotational (i.e., *angular* velocity $\boldsymbol{\omega}$) and translational components (i.e., *linear* velocity $\boldsymbol{v}$).

[7]Similarly, the inner product on the Lie group SE(3) essentially acts on elements of its Lie algebra $\mathfrak{se}(3)$.

[8]$\boldsymbol{v}_1^\top$ is a 3-dim row vector and $\boldsymbol{v}_2$ is a 3-dim column vector, so $\boldsymbol{v}_1^\top \boldsymbol{v}_2$ is a $1 \times 1$ matrix, and $\mathrm{tr}(\boldsymbol{v}_1^\top \boldsymbol{v}_2) = \boldsymbol{v}_1^\top \boldsymbol{v}_2$.

## G ADDITIONAL DETAILS ABOUT PROTEIN BACKBONE GENERATION

### G.1 PROTEIN BACKBONE PARAMETERIZATION

Here our protein backbone parameterization follows the seminal work of AlphaFold2 (Jumper et al., 2021) in that we associate a rigid-body *frame* with each residue in the amino acid sequence. Specifically, an $N$ residue backbone is parameterized by a collection of $N$ orientation preserving rigid transformations, i.e., *frames*, and each frame maps from fixed coordinates of four heavy atoms $N^*, C_\alpha^*, C^*, O^* \in \mathbb{R}^3$ centered at $C_\alpha^* = (0,0,0)$. Notably, each atom coordinate assumes chemically

idealized bond angles and lengths measured experimentally (Engh & Huber, 2012). Thus, residue $i \in [N]$ is denoted as an action of $T^{(i)}$ on idealized coordinates of the backbone main atoms $[N^{(i)}, C_\alpha^{(i)}, C^{(i)}] = T^{(i)} \cdot [N^*, C_\alpha^*, C^*]$, where $T^{(i)}$ is a member of the special Euclidean group SE(3), the set of orientation preserving rigid transformations in Euclidean space. Each $T^{(i)}$ can be formally decomposed into two components $T^{(i)} = (r^{(i)}, x^{(i)})$ where $r^{(i)} \in$ SO(3) is a $3 \times 3$ rotation matrix and $x^{(i)} \in \mathbb{R}^3$ represents a translation vector. And we collectively denote all $N$ frames as $\mathbf{T} = [T^{(1)}, \ldots, T^{(N)}] \in$ SE(3)$^N$. Moreover, to construct the backbone oxygen atom O, we rotate $O^*$ around the bond between $C_\alpha$ and C with an additional torsion angle $\psi \in$ SO(2). Figure 10 visually shows our backbone parameterization with frames.

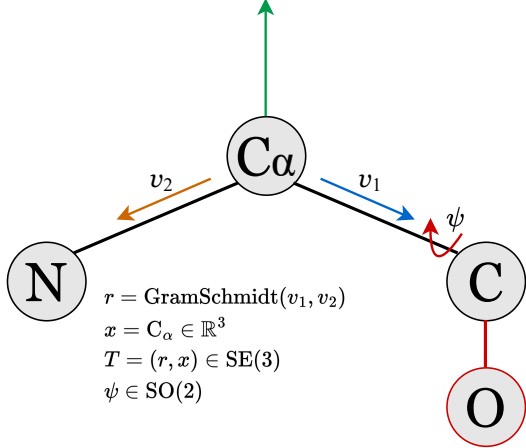

Figure 10: Protein parameterization with frames.

### G.2 CONVERSION BETWEEN COORDINATES AND FRAMES

As discussed above, $N^*, C_\alpha^*, C^*, O^*$ are idealized atom coordinates that assumes chemically idealized bond angles and lengths measured experimentally (Engh & Huber, 2012). However, these coordinates differ slightly per amino acid type. Here we uniformly take the idealized values of Alanine which are

$$N^* = (-0.525, 1.363, 0.0),$$
$$C_\alpha^* = (0.0, 0.0, 0.0),$$
$$C^* = (1.526, 0.0, 0.0),$$
$$O^* = (0.627, 1.062, 0.0).$$

Notably, these idealized values are derived with $C_\alpha^*$ as the origin. And using a central frame $T^{(i)}$, we can construct the realistic backbone main atoms of residue $i$ via $[N^{(i)}, C_\alpha^{(i)}, C^{(i)}] = T^{(i)} \cdot [N^*, C_\alpha^*, C^*]$. And the realistic backbone oxygen requires rotating a idealized oxygen around the $C - C_\alpha$ bond:

$$O^{(i)} = T^{(i)} \cdot T_{\text{psi}}^*(\psi^{(i)}) \cdot O^*, \tag{42}$$

where $\psi^{(i)}$ is a backbone torsion angle of residue $i$ and $T_{\text{psi}}^*(\psi^{(i)}) = (r_x(\psi^{(i)}), x_{\text{psi}})$ is a Euclidean transformation that maps the central frame $T^{(i)}$ to a new frame $T^{(i)} \cdot T_{\text{psi}}^*(\psi^{(i)})$ centered at C and rotated around the $x$-axis by $\psi^{(i)}$. Note $\psi^{(i)}$ is a tuple of two values describing a point on the unit circle, $\psi^{(i)} = [\psi_1^{(i)}, \psi_2^{(i)}]$ where $(\psi_1^{(i)})^2 + (\psi_2^{(i)})^2 = 1$. So $r_x(\psi^{(i)})$ and $x_{\text{psi}}$[9] can be expressed as

$$r_x(\psi^{(i)}) = \begin{pmatrix} 1 & 0 & 0 \\ 0 & \psi_1^{(i)} & -\psi_1^{(i)} \\ 0 & \psi_2^{(i)} & \psi_1^{(i)} \end{pmatrix}, \ x_{\text{psi}} = (1.526, 0.0, 0.0). \tag{43}$$

---

[9]The translation vector $x_{\text{psi}}$ transfers the oxygen atom with the rotation $r_x(\psi^{(i)})$ applied from the coordinate system with the origin $C_\alpha$ to the coordinate system with the origin C. Specifically, $T^{(i)}$ maps the ideal coordinate system (origin at $C_\alpha^*$) to the real coordinate system (origin at $C_\alpha^{(i)}$). In real space, the vector from $C_\alpha^{(i)}$ to $C^{(i)}$ is the same as the vector $(1.526, 0, 0)$ from $C_\alpha^*$ to $C^*$ in ideal space (because $T^{(i)}$ is a rigid-body transformation, preserving local distances and angles). So the translation vector $x_{\text{psi}}$ directly uses the ideal vector $(1.526, 0, 0)$.

To sum up, the mapping from frames to idealized coordinates, i.e., $\mathrm{frame2atom}$, is implemented by combining $[\mathrm{N}^{(i)}, \mathrm{C}_\alpha^{(i)}, \mathrm{C}^{(i)}] = T^{(i)} \cdot [\mathrm{N}^*, \mathrm{C}_\alpha^*, \mathrm{C}^*]$ and eq. (42), which can be formally expressed as

$$[\mathrm{N}^{(i)}, \mathrm{C}_\alpha^{(i)}, \mathrm{C}^{(i)}, \mathrm{O}^{(i)}] = \mathrm{frame2atom}(T^{(i)}, \psi^i). \tag{44}$$

We next introduce constructing rigid-body frames from atom coordinates, i.e., $\mathrm{atom2frame}$. Each frame can be obtained as described in the $\mathrm{rigidFrom3Points}$ algorithm in AF2 (Jumper et al., 2021):

---

**Algorithm 3:** Rigid from 3 points using the Gram-Schmidt process

---

**Data:** Coordinates of the three backbone atoms $\mathrm{N}^{(i)}, \mathrm{C}_\alpha^{(i)}, \mathrm{C}^{(i)}$ of residue $i$
**Result:** The rigid-body frame $T^{(i)}$ corresponding to residue $i$

1: **def** rigidFrom3Points($\mathrm{N}^{(i)}, \mathrm{C}_\alpha^{(i)}, \mathrm{C}^{(i)}$):             // $\mathrm{N}^{(i)}, \mathrm{C}_\alpha^{(i)}, \mathrm{C}^{(i)} \in \mathbb{R}^3$
2:    $\vec{v}_1 = \mathrm{C}^{(i)} - \mathrm{C}_\alpha^{(i)}$
3:    $\vec{v}_2 = \mathrm{N}^{(i)} - \mathrm{C}_\alpha^{(i)}$
4:    $\vec{e}_1 = \vec{v}_1 / \|\vec{v}_1\|$
5:    $\vec{u}_2 = \vec{v}_2 - \vec{e}_1(\vec{e}_1^\top \vec{v}_2)$
6:    $\vec{e}_2 = \vec{u}_2 / \|\vec{u}_2\|$
7:    $\vec{e}_3 = \vec{e}_1 \times \vec{e}_2$
8:    $r^{(i)} = \mathrm{concat}(\vec{e}_1, \vec{e}_2, \vec{e}_3)$                     // $r^{(i)} \in \mathbb{R}^{3 \times 3}$
9:    $x^{(i)} = \mathrm{C}_\alpha^{(i)}$
10: **return** $(r^{(i)}, x^{(i)})$

---

The conversion from coordinates to frames can be expressed as $T^{(i)} = \mathrm{atom2frame}(\mathrm{N}^{(i)}, \mathrm{C}_\alpha^{(i)}, \mathrm{C}^{(i)})$.

### G.3 FRAMEPRED ARCHITECTURE

**Overview of FramePred.** To predict the rigid-body frame for every protein residue, we utilize the FramePred architecture introduced in FrameDiff (Yim et al., 2023b) which performs iterative updates to the frames over a series of $L$ layers using a combination of *spatial* and *sequence* based attention modules. Specifically, $\mathbf{h}_\ell = [h_\ell^{(1)}, \cdots, h_\ell^{(N)}] \in \mathbb{R}^{N \times D_h}$ are node embeddings of the $\ell$-th layer where $h_\ell^{(i)}$ is the embedding for residue $i \in [N]$. And $\mathbf{z}_\ell \in \mathbb{R}^{N \times N \times D_z}$ are edge embeddings with $z_\ell^{(nm)}$ encoding the edge between residues $n$ and $m$. The frame of each residue at the $\ell$-th is denoted as $\mathbf{T}_\ell \in \mathrm{SE}(3)^N$. Unless stated otherwise, all instances of Multi-Layer Perceptrons (MLPs) use 3 Linear layers with biases, ReLU activation, and LayerNorm after the final layer. When FramePred is running, node embeddings $\mathbf{h}_\ell$ are first updated using Invariant Point Attention (IPA) (Jumper et al., 2021) with a skip connection. Before Transformer, the initial node embeddings $\mathbf{h}_0$ and post-IPA embeddings are concatenated. After transformer, we include a skip connection with post-IPA embeddings. The updated node embeddings $\mathbf{h}_{\ell+1}$ are then used to update edge embeddings $\mathbf{z}_{\ell+1}$ as well as predict frame updates $\mathbf{T}_{\ell+1}$. And so on to get the final frames $\mathbf{T}_L$ of all protein residues.

**Feature initialization.** Following the methodology established by Trippe et al. (2022), node and edge embeddings are initialized using a combination of residue indices and timestep information. Specifically, sinusoidal embeddings $\phi(\cdot)$ (Vaswani et al., 2017) are applied to these features, after which an MLP is used to project them into the desired embedding space. For residue $i \in [N]$, the initial node embedding at layer 0 incorporates the residue index $i$ and the diffusion timestep $t$, i.e., $h_0^{(i)} = \mathrm{MLP}([\phi(n), \phi(t)]) \in \mathbb{R}^{D_h}$ [10], where $D_h$ denotes the dimension of node embeddings. Moreover, for a residue pair $(n, m)$, the edge embedding $z_0^{(nm)}$ additionally includes the relative sequence distance $\phi(m - n)$ and a binned displacement feature derived from self-conditioned $\mathrm{C}_\alpha$ coordinates $\phi(\mathrm{disp}_{sc}^{(nm)})$. The initial edge embeddings can be formally expressed as

$$z_0^{(nm)} = \mathrm{MLP}([\phi(n), \phi(m), \phi(m - n), \phi(t), \phi(\mathrm{disp}_{sc}^{(nm)})]) \in \mathbb{R}^{D_z}, \tag{45}$$

---

[10]Here we stipulate that superscripts without parentheses are used to refer to time step, superscript numbers within parentheses refer to residue indices, and subscripts refer to variable names (layer indices in most cases).

where $D_z$ denotes the dimension of edge embeddings and $\text{disp}_{sc}$ is the self-conditioning of predicted $C_\alpha$ displacements. Specifically, let $\hat{x}_{sc}$ be the $C_\alpha$ coordinates (in Å) predicted during self-conditioning. And to prevent over-reliance on self-conditioned outputs, we set $\hat{x}_{sc} = 0$ with 50% probability during training. The binned displacement between residues $n$ and $m$ is formally expressed as

$$\text{disp}_{sc}^{(mn)} = \sum_{i=1}^{N_{\text{bins}}} \mathbb{1}\{|\hat{x}_{sc}^{(n)} - \hat{x}_{sc}^{(m)}| < \nu_i\}, \tag{46}$$

where $\nu_1, \cdots, \nu_{N_{\text{bins}}} = \text{Linspace}(0, 20)^{11}$ are equally spaced intervals between 0 and 20 angstroms.

To construct initial frames, $C_\alpha$ coordinates are first zero-centered and all backbone coordinates $(N, C_\alpha, C, O)$ are scaled to nanometers as done in AF2 (Jumper et al., 2021) by multiplying coordinates by $1/10$. We then construct initial frames for each protein residue $i$ as

$$T^{0,(i)} = (r^{0,(i)}, x^{0,(i)}) = \text{atom2frame}(N^{(i)}, C_\alpha^{(i)}, C^{(i)}). \tag{47}$$

**Node update.** IPA is first introduced in AF2 (Jumper et al., 2021) and we apply this algorithm without modifications. And Transformer is also used without modification from Vaswani et al. (2017). Node update is formally represented as follows, including specific operations and data dimensions.

$$\mathbf{h}_{\text{ipa}} = \text{LayerNorm}(\text{IPA}(\mathbf{h}_\ell, \mathbf{z}_\ell, \mathbf{T}_\ell) + \mathbf{h}_\ell) \in \mathbb{R}^{N, D_h}$$

$$\mathbf{h}_{\text{skip}} = \text{Linear}(\mathbf{h}_0) \in \mathbb{R}^{N, D_{\text{skip}}}$$

$$\mathbf{h}_{\text{in}} = \text{concat}(\mathbf{h}_{\text{ipa}}, \mathbf{h}_{\text{skip}}) \in \mathbb{R}^{N, (D_{\text{skip}} + D_h)}$$

$$\mathbf{h}_{\text{trans}} = \text{Transformer}(\mathbf{h}_{\text{in}}) \in \mathbb{R}^{N, (D_{\text{skip}} + D_h)}$$

$$\mathbf{h}_{\text{out}} = \text{Linear}(\mathbf{h}_{\text{trans}}) + \mathbf{h}_\ell \in \mathbb{R}^{N, D_h}$$

$$\mathbf{h}_{\ell+1} = \text{MLP}(\mathbf{h}_{\text{out}}) \in \mathbb{R}^{N, D_h}$$

**Edge update.** Each directed edge is updated through an MLP of the current edge and the embeddings of the source and target nodes. Edge update is also formally expressed as follows.

$$\mathbf{h}_{\text{down}} = \text{Linear}(\mathbf{h}_{\ell+1}) \in \mathbb{R}^{N, D_h/2}$$

$$z_{\text{in}}^{(nm)} = \text{concat}(h_{\text{down}}^{(n)}, h_{\text{down}}^{(m)}, z_\ell^{(nm)}) \in \mathbb{R}^{N, (D_h + D_z)}$$

$$\mathbf{z}_{\ell+1} = \text{LayerNorm}(\text{MLP}(\mathbf{z}_{\text{in}})) \in \mathbb{R}^{N, N, D_z}$$

Notably, in the first line, node embeddings are first projected down to half the dimension.

**Backbone update.** As for the backbone update, we follow the BackboneUpdate algorithm proposed in AF2 (Jumper et al., 2021) and present its specific operations in detail as follows.

$$b^i, c^i, d^i, x_{\text{update}}^{(i)} = \text{Linear}(h_\ell^i)$$

$$(a^i, b^i, c^i, d^i) = (1, b^i, c^i, d^i)/\sqrt{1 + b^i + c^i + d^i}$$

$$r_{\text{update}}^{(i)} = \begin{pmatrix} (a^i)^2 + (b^i)^2 - (c^i)^2 - (d^i)^2 & 2b^i c^i - 2a^i d^i & 2b^i d^i + 2a^i c^i \\ 2b^i c^i + 2a^i d^i & (a^i)^2 - (b^i)^2 + (c^i)^2 - (d^i)^2 & 2c^i d^i - 2a^i b^i \\ 2b^i d^i - 2a^i c^i & 2c^i d^i + 2a^i b^i & (a^i)^2 - (b^i)^2 - (c^i)^2 + (d^i)^2 \end{pmatrix}$$

$$T_{\text{update}}^{(i)} = (r_{\text{update}}^{(i)}, x_{\text{update}}^{(i)})$$

$$T_{\ell+1}^{(i)} = T_\ell^{(i)} \cdot T_{\text{update}}^{(i)}$$

where $b^i, c^i, d^i \in \mathbb{R}^{12}$, $r_{\text{update}}^{(i)} \in \text{SO}(3)$, and $x_{\text{update}}^{(i)} \in \mathbb{R}^3$. Here we first constructs a normalized quaternion (2nd line) and then convert it into a valid rotation matrix (3rd line).

**Torsion Prediction.** We still follow AF2 (Jumper et al., 2021) to predict the torsion angle $\psi$.

$$\mathbf{h}_{\text{psi}} = \text{MLP}(\mathbf{h}_L) \in \mathbb{R}^{N, D_h}$$

$$\boldsymbol{\psi}_{\text{unnormalized}} = \text{Linear}(\mathbf{h}_{\text{psi}} + \mathbf{h}_L) \in \text{SO}(2)^N$$

$$\hat{\boldsymbol{\psi}} = \boldsymbol{\psi}_{\text{unnormalized}}/||\boldsymbol{\psi}_{\text{unnormalized}}|| \in \text{SO}(2)^N$$

---

[11]In our experiments we set $N_{\text{bins}} = 22$.

[12]Due to space limitations, we use superscripts without parentheses instead of superscripts with parentheses.

# H COMPLETE PROTEIN EXPERIMENTS

## H.1 EXPERIMENTAL SETUP

**Training.**  To verify the effectiveness of Momentum Flow on the protein monomer generation task, following GENIE (Lin & AlQuraishi, 2023) and FrameFlow (Yim et al., 2023a), we train it on the SCOPe dataset (Chandonia et al., 2022) with proteins below length 128 for a total of 3, 938 examples. Our model is trained for 10 hours on two NVIDIA A800-80G GPUs using the batching strategy from FrameDiff (Yim et al., 2023b) of combining proteins with the same length into the same batch to remove extraneous padding and we take the optimal 3 checkpoints for evaluation. We use the Adam (Adam et al., 2014) optimizer with a learning rate of 0.0001, $\beta_1 = 0.9$, and $\beta_2 = 0.999$.

**Metrics.**  During evaluation, we sample 10 backbones for every length between 60 and 300 then use ProteinMPNN (Dauparas et al., 2022) to design 8 sequences for each backbone. Notably, the upper limit of 300 here differs from the upper limit of 128 during training. We increase the upper limit during evaluation to show the generalization of our model in generating long sequence proteins.

The assessment of generated protein structures employs complementary metrics evaluating distinct aspects of protein quality. ***Accuracy*** is quantified by comparing predictions against native structures in PDB using scRMSD for atomic-level precision and TM-score (referred to as pdbTM) for global topological fidelity, with a TM-score $> 0.5$ indicating a correct fold. Intrinsic structural plausibility, a proxy for *designability*, is assessed using ***confidence*** estimates from deep learning models: pLDDT reports per-residue local reliability, while pAE evaluates the self-consistency of long-range inter-residue distances. Moreover, scTM and scRMSD also serve as the fundamental distance measures for quantifying *novelty* (against known structures in PDB) and *diversity* (within a generated ensemble).

**Baselines.**  We compare our results to GENIE (Lin & AlQuraishi, 2023) and FrameFlow (Yim et al., 2023a), a diffusion model and a rectified flow model for protein backbone generation, respectively, that do not rely on a pre-trained folding network. We retrain both models according to their default recommended settings. Our baselines are intended to demonstrate tradeoffs in efficiency and diversity.

## H.2 HYPERPARAMETERS

**Neural network hyperparameters.**

$$\text{Global parameters}: \quad D_h = 256 \quad D_z = 128 \quad D_{\text{skip}} = 64 \quad L = 4$$
$$\text{IPA parameters}: \quad \text{heahs} = 8 \quad \text{query points} = 8 \quad \text{value points} = 12$$
$$\text{Transformer parameters}: \quad \text{heads} = 4 \quad \text{layers} = 2$$

With these parameters, our neural network has 17, 446, 190 trainable weights.

**SDE parameters.**

$$\text{Translations}: \quad \text{schedule} = \text{linear} \quad \beta_{\min} = 0.1 \quad \beta_{\max} = 20$$
$$\text{Rotations}: \quad \text{schedule} = \text{logarithmic} \quad \beta_{\min} = 0.1 \quad \beta_{\max} = 1.5$$

## H.3 FURTHER EXPERIMENTAL ANALYSIS

We use the Euler-Maruyama integrator for SDE sampling and the Euler integrator for ODE sampling. The number of integration timesteps for all methods is set to 100. Quantitative results are shown in Table 3. We use SDE sampling for GENIE (Lin & AlQuraishi, 2023) and ODE sampling for FrameFlow (Yim et al., 2023a) since these are the methods used in their respective papers.

As illustrated in Table 3, both GENIE (Lin & AlQuraishi, 2023) and FrameFlow (Yim et al., 2023a) exhibit limitations—either in sampling fidelity or structural plausibility—our model consistently outperforms them across all four metrics. Our momentum flow significantly improves structural accuracy, as evidenced by the highest scTM (0.47) and lowest scRMSD (8.05), indicating generated backbones more closely resemble native-like folds. Meanwhile, our momentum flow enhances designability, reflected in the highest pLDDT (70.09) and lowest pAE (9.50), suggesting superior local and global structural self-consistency without relying on ground-truth alignment.

While GENIE has less parameters (4.1M) than FrameFlow/Momentum Flow (17.4M), i.e., the FramePred architecture introduced in App. G.3, it uses expensive triangle updates (Jumper et al., 2021) that requires high memory cost and greater compute for each forward call. Sampling a length 100 protein with 100 timesteps on an NVIDIA A800-80G GPU takes GENIE around 10 seconds while for FrameFlow/Momentum Flow sampling with 100 timesteps takes around 4 seconds.

In conclusion, our Momentum Flow achieves a more favorable efficient-diverse trade-off, where high-quality and various protein backbone samples can be generated with reduced computational overhead——a critical advantage for practical protein design applications.

## I   LLM USAGE STATEMENT

During the preparation of this manuscript, we employed GPT-5 exclusively for language refinement. The model was instructed to improve grammar, clarity, and readability while preserving the original meaning of the content.

