# OpenReview forum: "Flow Diverse and Efficient: Learning Momentum Flow Matching via Stochastic Velocity Field Sampling"
_ICLR.cc/2026/Conference — ICLR 2026 Conference Withdrawn Submission_

### Official Review · Reviewer_6CbM · 2025-10-26

**Soundness:** 3
**Presentation:** 3
**Contribution:** 3
**Rating:** 2
**Confidence:** 4

**Summary:**

In rectified flow models, the straight-line mapping between noisy and clean data distributions offers a fast and intuitive optimization path. However, this formulation raises two issues: limited sample diversity and insufficient multi-scale noise modeling.

To address these, the paper proposes Discretized-RF, a momentum-based flow matching model that incorporates both previous and random velocity components at each step. It discretizes the straight trajectory into a sequence of variable velocity sub-paths, referred to as *momentum fields*, thereby expanding the search space. Unlike prior approaches that add noise directly to $x$, this method injects noise into the velocity field $v$ of each sub-path, altering its direction to enhance diversity and improve multi-scale noise modeling.

**Strengths:**

- The paper extends constant velocity field models to acceleration (momentum) field models, aiming for a better trade-off between sampling efficiency and diversity. It introduces a momentum-driven flow model that discretizes the straight path into variable velocity sub-paths, achieving trajectories that are more deterministic near the data distribution and more stochastic near the noise distribution—balancing efficiency and diversity without losing the simplicity of straight-line transport.
- The proposed segmented straight-line sampling improves multi-scale noise modeling, offering a finer approximation of noise addition and denoising compared to standard rectified flows.
- The method also shows adaptability to SE(3) protein generation tasks, extending the approach beyond image domains to non-Euclidean manifolds.

**Weaknesses:**

- Some baseline comparisons appear limited; in Table 2, the proposed method does not consistently outperform existing rectified flow baselines across all metrics or settings, making it unclear when and how much improvement is achieved. Additional comparisons with other *training-improved rectified flow methods* would help contextualize the results.
- The empirical gains are relatively modest, and the trade-off between sampling efficiency and diversity is not clearly characterized in Figure 4. The observed trends are weak, suggesting that further analysis is needed to substantiate the claimed benefits.

**Questions:**

- For Figure 4, it would be useful to present a Pareto plot showing (Recall, FID) scatter points with $\gamma$ values color-coded, to visualize how $\gamma$  influences the Pareto frontier and the balance between quality and diversity.
- The authors should quantitatively compare performance gains against other approaches that improve rectified flow training—for instance,
    - Improving the Training of Rectified Flows (Lee et al., NeurIPS 2024), and
    - SlimFlow: Training Smaller One-Step Diffusion Models with Rectified Flow (Zhu et al., ECCV 2024).

---

### Official Review · Reviewer_d1Dt · 2025-10-29

**Soundness:** 2
**Presentation:** 2
**Contribution:** 2
**Rating:** 4
**Confidence:** 4

**Summary:**

The paper proposed momentum flow matching, a novel generative model that combines diffusion model and rectified flow. In the forward process, data is perturbed via diffusion process, and rectified flow is trained to match velocity between anchor pairs on a smaller time scale. At inference time, repeated ODE integration on the small time scale leads to the final generation. The method improves generation diversity compared to rectified flow, and efficiency compared to diffusion model. It's effectiveness is demonstrated through experiments on high-resolution image and protein backbone generation.

**Strengths:**

1. Momentum flow matching finds a middle ground between rectified flow with straight path and diffusion model with noisy path. Aiming for a piecewise straight path on a multi-scale noise model, momentum flow matching enhances the generation diversity of rectified flow and improves the inference speed of diffusion model.
2. The protein backbone generation experiments, i.e. generation on SE(3), successfully supported the author's argument.

**Weaknesses:**

1. Lack of motivation: The paper aims to tackle two concerns of rectified flow, namely generation diversity and multi-scale modeling. The first concern, according to the authors, is due to the fact that rectified flow is too "straight". However the general form of stochastic interpolants [1], a concurrent work of rectified flow, has introduced methods of matching velocities of noisy trajectories, which solves the issue of straightness in flow matching. Comparing merely to rectified flow without mentioning stochastic interpolants substantially weakens the motivation. For the second concern, it is never clearly explained in the paper why multi-scale modeling is essential.
2. Lack of novelty: Momentum flow matching essentially takes two consecutive points on a diffusion trajectory and learns a rectified flow to match them, resulting in a piecewise-straight diffusion trajectory. Without further insights, the method seems to be a concatenation of diffusion model and rectified flow. The authors should consider explaining the rationale and advantage of modeling the momentum, and compare with methods of similar idea [2][4].
3. Experiment: FID score in Table 1 & 2 seems unreasonably high across all experiments. Visual qualities of the generated images can't match state-of-the-art. For the baseline NanoFlow [3], a paper that concerns parallel computing of large language models, there seems to be no connection to any of the experiments carried out in the momentum flow matching paper.

[1] Albergo, Michael S., Nicholas M. Boffi, and Eric Vanden-Eijnden. "Stochastic interpolants: A unifying framework for flows and diffusions." arXiv preprint arXiv:2303.08797 (2023).

[2] Zhang, Yichi, et al. "Towards Hierarchical Rectified Flow." The Thirteenth International Conference on Learning Representations.

[3] Zhu, Kan, et al. "NanoFlow: Towards Optimal Large Language Model Serving Throughput." CoRR (2024).

[4] Yan, Hanshu, et al. "Perflow: Piecewise rectified flow as universal plug-and-play accelerator." Advances in Neural Information Processing Systems 37 (2024): 78630-78652.

**Questions:**

Questions follows from the weakness.
1. Is it possible that the authors compare with stochastic interpolants for generation diversity?
2. Is it possible to justify why multi-scale noisy scheduling is advantageous?
3. Is it possible to compare with Hierarchical Rectified Flow?
4. Can the authors explain why FID of rectified flow is much higher than in the original RF paper?
5. Can the authors explain how they used NanoFlow to generate images?
6. Can the authors add a step length \Delta_t to all ODE discretization? The current time lags are all of unit length which is hardly reasonable.
7. Can the authors fix typos in Algorithms 2 Step 4?

---

### Official Review · Reviewer_GKuD · 2025-10-31

**Soundness:** 3
**Presentation:** 2
**Contribution:** 2
**Rating:** 4
**Confidence:** 4

**Summary:**

This paper proposes Momentum Flow Matching (MFM), a discretized extension of Rectified Flow that injects stochastic momentum noise into the velocity field to balance sampling efficiency and diversity. Experiments on image and protein generation show improved FID/recall trade-offs and adaptability to SE(3) manifolds.

**Strengths:**

- The paper identifies a meaningful goal: improving the diversity-efficiency trade-off in rectified flow models through stochastic velocity perturbation.
- The momentum formulation provides an intuitive physical interpretation that connects rectified flows and stochastic diffusion dynamics under a unified velocity-based view.
- Experiments show consistent quantitative improvements.

**Weaknesses:**

- The core idea is extremely similar to PeRFlow [1].
- The algorithmic description lacks rigor and clarity. It is not explained how $(z_{t-1},z_t)$ are drawn, nor how the ODE is solved. Do the authors use Algorithm 1 and do the simulation? How do you solve ODE $\frac{dz_t}{dt}=u_{\theta}(z_t^m,m)$ with $z_0\sim\pi_0$. It appears that integration should be performed over $m$ from 0 to 1.
- There seems to be no theoretical guarantee of marginal consistency. Can the authors show that integrating the velocity yields the correct marginal at each time $t$?
- Experimental baselines are poorly chosen. The Nanoflow seems irrelevant. The core idea is similar to PeRFlow [1] and HRF [2]. You should discuss and compare with the baselines addressing similar problems.
- The qualitative results show visibly degraded image quality when the number of function evaluations (NFE) is small. This undermines the claim of improved sampling efficiency, since high visual fidelity still requires a relatively large number of steps.

[1] Yan et al. PeRFlow: Piecewise Rectified Flow as Universal Plug-and-Play Accelerator.
[2] Zhang et al. Towards Hierarchical Rectified Flow. ICLR 2025.

**Questions:**

See weaknesses.

---

### Official Review · Reviewer_tF4D · 2025-11-01

**Soundness:** 2
**Presentation:** 2
**Contribution:** 2
**Rating:** 2
**Confidence:** 4

**Summary:**

This paper introduces Momentum Flow Matching (also called Discretized-RF), which discretizes the straight path of flow matching into a series of sub-paths with variable velocity fields. The velocity along this path evolves stochastically via a momentum-like update, making the transport more random near the noise distribution for diversity and more deterministic near the data distribution for efficiency. Experiments are conducted on image and protein generation tasks.

**Strengths:**

1. The paper attempts to convert ODE-based flow matching into SDE-based formulation (similar to DDPM) to improve diversity. The motivation is reasonable.
2. Figures 1 and 2 provide clear visual intuitions of the method.

**Weaknesses:**

1. Converting flow matching into stochastic differential equation form is not novel, as there already exist methods [1] that do this. The paper lacks both theoretical and empirical comparisons with these existing approaches.

2. The FID results on CIFAR-10 (Table 1) and ImageNet-64 (Table 2) are significantly worse than recent published results. For instance, the results are worse than those shown in Figure 2 (ODE-based) and Figure 4 (SDE-based) of [2].

3. Table 1 only reports FID, which cannot support the paper's claim of improving diversity.

[1] Liu, Jie, et al. "Flow-grpo: Training flow matching models via online rl." arXiv preprint arXiv:2505.05470 (2025).

[2] Karras, Tero, et al. "Elucidating the design space of diffusion-based generative models." Advances in neural information processing systems 35 (2022): 26565-26577.

**Questions:**

1. How does this method compare to standard flow matching when NFE is sufficiently large? For example, in Table 1 of [1], 1-Rectified Flow achieves FID=2.58, Recall=0.57, IS=9.60 at NFE=127.

2. Is the "50.26" in Table 1 a reproduced result of RectifiedFlow or taken from a specific table in the original RectifiedFlow paper [1]?

[1] Liu, Xingchao, Chengyue Gong, et al. "Flow straight and fast: Learning to generate and transfer data with rectified flow." In The Eleventh International Conference on Learning Representations, 2022.

---

### Note · Authors · 2025-11-12

I have read and agree with the venue's withdrawal policy on behalf of myself and my co-authors.